# Expansion and loss of sperm nuclear basic protein genes in *Drosophila* correspond with genetic conflicts between sex chromosomes

**Ching-Ho Chang[1]\*, Isabel Mejia Natividad[1,2], Harmit S Malik[1,2]**

[1]Division of Basic Sciences, Fred Hutchinson Cancer Center, Seattle, United States; [2]Howard Hughes Medical Institute, Fred Hutchinson Cancer Center, Seattle, United States

**Abstract** Many animal species employ sperm nuclear basic proteins (SNBPs) or protamines to package sperm genomes tightly. SNBPs vary across animal lineages and evolve rapidly in mammals. We used a phylogenomic approach to investigate SNBP diversification in *Drosophila* species. We found that most SNBP genes in *Drosophila melanogaster* evolve under positive selection except for genes essential for male fertility. Unexpectedly, evolutionarily young SNBP genes are more likely to be critical for fertility than ancient, conserved SNBP genes. For example, *CG30056* is dispensable for male fertility despite being one of three SNBP genes universally retained in *Drosophila* species. We found 19 independent SNBP gene amplification events that occurred preferentially on sex chromosomes. Conversely, the *montium* group of *Drosophila* species lost otherwise-conserved SNBP genes, coincident with an X-Y chromosomal fusion. Furthermore, SNBP genes that became linked to sex chromosomes via chromosomal fusions were more likely to degenerate or relocate back to autosomes. We hypothesize that autosomal SNBP genes suppress meiotic drive, whereas sex-chromosomal SNBP expansions lead to meiotic drive. X-Y fusions in the *montium* group render autosomal SNBPs dispensable by making X-versus-Y meiotic drive obsolete or costly. Thus, genetic conflicts between sex chromosomes may drive SNBP rapid evolution during spermatogenesis in *Drosophila* species.

**\*For correspondence:**
hilynano@gmail.com

**Competing interest:** The authors declare that no competing interests exist.

## Editor's evaluation

Chang et al. used a previously published set of highly contiguous genomes to infer the drivers of the evolution of sperm nuclear basic proteins and find several instances of gene duplication mainly occurring in sex chromosomes. Moreover, they provide a genetic characterization of one such protein (CG30056). The paper was initially reviewed by experts in the field through Review Commons. The three reviewers were enthusiastic about the potential of the paper and made a set of suggestions to make the paper stronger. The authors incorporated the suggestions when relevant, added a new (and relevant) experiment, edited the manuscript as requested, and clarified some instances that could be further developed.

## Introduction

Chromatin plays a critical role in organizing genomes and regulating gene expression. Histones are the primary protein components of chromatin in most eukaryotes. Due to their conserved, essential functions, most histone proteins are ancient and subject to strong evolutionary constraints, although

**eLife digest** In sperm, DNA is packaged more tightly than in other cells thanks to small proteins called 'sperm nuclear basic proteins' (SNBPs), also called protamines in mammals. SNBPs are important for sperm to develop properly and correctly perform their role during fertilization. Although the evolution of SNBPs has been studied in mammals, these proteins have not been as thoroughly examined in invertebrates.

Chang et al. took advantage of the availability of high-quality sequences for the genomes of 78 species of *Drosophila* flies to investigate the evolution of the genes that code for SNBPs in these flies. The results showed that, just like in mammals, in *Drosophila* the protein sequences of SNBPs evolve rapidly. However, unlike mammals, Chang et al. also found that *Drosophila* species frequently gained and lost genes coding for SNBPs.

Interestingly, the 'older' genes (genes that appeared earlier in evolution) that code for SNBPs are not essential for reproduction in the fruit fly *Drosophila melanogaster*. This is an unexpected finding because older genes usually have essential roles for survival and reproduction, which require them to be passed on to the next generation and remain in the genome. In contrast, younger SNBP genes that had appeared more recently and were not shared between different species of *Drosophila* were often essential for fertility.

These results, combined with other observations about where SNBP genes are located in the genome, led Chang et al. to hypothesize that SNBPs present in sex chromosomes act as 'meiotic drivers' while those on other chromosomes (known as autosomes) suppress meiotic drive. In other words, SNBP genes present in the sex chromosomes may be responsible for killing sister sperm cells that do not carry those genes, while SNBP genes that are not located on sex chromosomes may suppress this activity. This is of particular interest because it indicates that SNBPs are involved in genetic conflicts between the two sex chromosomes: sperm that carry SNBPs on the X chromosome may kill sperm with a Y chromosome, and vice versa.

The results of Chang et al. shed light on the mysterious evolution of SNBPs in *Drosophila* flies. Although previous hypotheses regarding the rapid evolution of SNBPs evolution have focused on their role in genome packaging, this new analysis suggests that much of the evolutionary change is likely driven by genetic conflicts between sex chromosomes.

there are distinct exceptions among histone variants (*Raman et al., 2022*; *Molaro et al., 2020*; *Talbert and Henikoff, 2021*). Many animal species replace most histones with sperm nuclear basic proteins (SNBPs) to package their genomes more tightly into tiny sperm heads during spermatogenesis (*Sassone-Corsi, 2002*; *Ward and Coffey, 1991*). Like histones, most SNBPs are small (<200 amino acids) and positively charged. Many SNBPs also contain a high proportion of lysine, arginine, and cysteine residues, which form disulfide bridges to further condense DNA within sperm heads (*Török et al., 2016*; *Eirín-López and Ausió, 2009*). As a result of their tighter DNA packaging, SNBPs can reduce the size of the sperm nuclei by 20–200-fold compared to histone-enriched nuclei (*Brewer et al., 1999*). Based on their role in condensing sperm nuclei, the prevailing hypothesis is that sexual selection for competitive sperm shapes led to the evolutionary origins of SNBP genes in most animal taxa (*Lüke et al., 2014*; *Lüpold et al., 2016*).

SNBPs have been most well-studied in mammals (*Balhorn, 2007*). Mammalian SNBPs in mature sperm include protamine 1 (*PRM1*) and protamine 2 (*PRM2*), which are encoded in an autosomal gene cluster that includes Transition Protein 2 (*TNP2*) and protamine 3 (*PRM3*) (*Martin-Coello et al., 2011*). Although these four genes share moderate homology, *TPN2* is only expressed during the histone-to-protamine transition (*Nayernia et al., 1996*), whereas PRM3 only localizes to the cytoplasm of elongated spermatids (*Martin-Coello et al., 2011*). Both *PRM1* and *PRM2* are essential for fertility in humans and mice; their expression levels directly affect sperm quality (*Balhorn, 2007*). Loss of *PRM1* or *PRM2* leads to defects in sperm head morphology and fertility in mice and humans (*Cho et al., 2003*; *Cho et al., 2001*). Yet, *PRM2* has undergone pseudogenization in bulls (*Balhorn, 2007*). Thus, even SNBPs essential for male fertility can be subject to evolutionary turnover in some species.

Although SNBPs play a similar genome-packaging role to histones, they differ dramatically from histones in their evolutionary origins and trajectories. Whereas histones have ancient origins, SNBPs

were independently derived from different ancestral proteins across taxa (*Eirín-López and Ausió, 2009*; *Reynolds and Wolfe, 1984*; *Török and Gornik, 2018*). For example, SNBPs arose from linker histone H1 gene variants in liverworts and tunicates (*Lewis et al., 2004*; *D'Ippolito et al., 2019*), whereas they arose from histone H2B gene variants in cnidarians and echinoderms (*Török et al., 2016*; *Green and Poccia, 1988*; *Eirín-López et al., 2006*). SNBPs in other animals lack apparent homology to other existing proteins, obscuring their evolutionary origins (*Eirín-López et al., 2006*). In addition to their convergent evolution and turnover, SNBPs differ dramatically from histones in their evolutionary rates of amino acids. For example, *PRM1* and *PRM2* are among the most rapidly evolving protein-coding genes encoded in mammalian genomes (*Saperas and Ausió, 2013*) and evolve under positive selection in many lineages (*Lüke et al., 2014*; *Wyckoff and Wang, 2000*). The positive selection of SNBPs results in changes to their amino acid composition. For example, the arginine content of PRM1 is partially correlated across species with sperm head length, which may reflect the selective pressures of sperm competition (*Lüke et al., 2016b*). The rapid evolution of *PRM1* and *PRM2* is consistent with sexual selection on sperm heads driving SNBP origins and rapid evolution in mammals (*Wyckoff and Wang, 2000*; *Lüke et al., 2016b*; *Lüke et al., 2016a*), although this hypothesis has yet to be experimentally tested. Moreover, the evolutionary trajectories of SNBP genes and their underlying causes have not been deeply investigated outside mammals.

*Drosophila* species provide an excellent model to study SNBP function and evolution due to the ease of genetic manipulations and sperm biology characterization, and the availability of high-quality genome sequences from many closely related species. Previous studies have shown that *Drosophila* SNBPs independently arose from proteins encoding high mobility group box (HMG-box) DNA-binding proteins (*Gärtner et al., 2015*); thus, they have distinct origins and likely functions from mammalian protamines. Five HMG-box SNBP genes have been previously identified in *Drosophila melanogaster*: *ProtA, ProtB, ddbt, Mst77F,* and *Prtl99C* (*Tirmarche et al., 2014*; *Kimura and Loppin, 2016*; *Jayaramaiah Raja and Renkawitz-Pohl, 2005*; *Eren-Ghiani et al., 2015*; *Yamaki et al., 2016*). Each of these five SNBPs is incorporated into nuclei independently of each other, suggesting that they play distinct roles in sperm formation (*Kimura and Loppin, 2016*; *Eren-Ghiani et al., 2015*; *Rathke et al., 2010*). Based on the common HMG-box motifs found in these five SNBPs, 10 other male-specific proteins with the same motif were later identified in *D. melanogaster* (*Gärtner et al., 2015*; *Eren-Ghiani et al., 2015*), and 4 of them were later shown to be enriched in sperm nuclei (*Gärtner et al., 2015*). Other proteins without any HMG-box are also demonstrated to locate in sperm nuclei, but it is unclear whether they bind to DNA (*Rivard et al., 2021*; *Hempel et al., 2006*; *Harhangi et al., 1999*).

Recent studies in *Drosophila* have suggested an alternate hypothesis other than sperm competition—meiotic drive and its suppression—to explain the rapid diversification and innovation of SNBP-like proteins (*Vedanayagam et al., 2021*; *Muirhead and Presgraves, 2021*). Meiotic drivers are selfish elements that can bias their transmission via hijacking meiosis or post-meiosis processes, e.g., killing sperm that do not carry the driver. These genetic drivers exist in widespread lineages, including plants, animals, and fungi (*Courret et al., 2019*). One of the first identified drivers is *Segregation Distorter* in *D. melanogaster*, whose drive strength can be further enhanced by the knockdown of *ProtA/ProtB* genes (*Gingell and McLean, 2020*). Thus, *ProtA/ProtB* serve as suppressors of meiotic drive through an unknown mechanism. The second piece of evidence emerged from studies of *Distorter on the X (Dox)*, an X-chromosomal driver in *Drosophila simulans* (*Tao et al., 2007a*). *Dox* emerged via the stepwise acquisition of multiple gene segments, mostly from *ProtA/ProtB*. *Dox* produces chromosome condensation defects in Y chromosome-containing sperm during spermatogenesis, ultimately leading to X-chromosomal bias among functional sperm and sex-ratio bias in resulting progeny (*Tao et al., 2007a*; *Faulhaber, 1967*; *Tao et al., 2007b*). In *D. simulans* and sister species, *Dox*-like genes have amplified and diversified on the X chromosome in an escalating battle between X and Y chromosomes for transmission through the male germline (*Vedanayagam et al., 2021*; *Muirhead and Presgraves, 2021*). Thus, genetic conflicts between sex chromosomes and their suppression of those conflicts could provide an alternate explanation for the recurrent diversification of SNBP genes in *Drosophila* species.

Here, we systemically explored the evolution of SNBP genes via a detailed phylogenomic analysis across *Drosophila* species. We found that SNBP genes are rapidly evolving, and most of them are under positive selection in *Drosophila*, like in mammals. Thus, the rapid sequence changes of SNBP genes are common to many animal taxa. Interestingly, we found an inverse relationship between age

and essentiality; young SNBPs are essential for male fertility in *D. melanogaster*, whereas ancient, conserved SNBPs are not. Moreover, SNBP genes essential for male fertility in *D. melanogaster* are frequently lost in other *Drosophila* species. Unexpectedly, we found 19 independent amplification events from eight different SNBP genes on either X or Y chromosomes in *Drosophila* species. Conversely, species with reduced conflicts between sex chromosomes due to chromosomal fusions do not undergo SNBP amplification, but instead lose SNBP genes. Thus, we conclude that rapid diversification of SNBP genes might be largely driven by genetic conflicts between sex chromosomes in *Drosophila*.

## Results

### SNBP genes in *Drosophila* species

To study SNBP evolution in *Drosophila* species, we performed a detailed survey of all testis-specific genes encoding HMG boxes in *D. melanogaster*. Our survey did not reveal any additional genes beyond the 15 previously identified autosomal SNBP genes, which function at different stages of spermatogenesis (*Table 1*). For example, *CG14835, ProtA, ProtB, Mst77F, Prtl99C,* and *ddbt* all encode SNBP proteins present in the mature sperm head (*Jayaramaiah Raja and Renkawitz-Pohl, 2005*; *Eren-Ghiani et al., 2015*; *Yamaki et al., 2016*). In contrast, *Tpl94D, tHMG-1, tHMG-2,* and *CG30356* encode transition SNBP proteins during the transition between histone and protamines but are not retained in mature sperm (*Gärtner et al., 2015*; *Rathke et al., 2007*). The five remaining SNBP genes (*Mst33A, CG30056, CG31010, CG34269,* and *CG42355*) remain cytologically uncharacterized (*Eren-Ghiani et al., 2015*). Using single-cell transcriptomic data (*Witt et al., 2021*), we confirmed that all candidate SNBP genes are transcribed in male germline cells, with the highest level of expression of most SNBP genes occurring in late spermatocytes. The only exceptions are *CG34269*, which is transcribed earlier in late spermatogonia, and *CG30056*, which is transcribed later in late spermatids (*Figure 1—figure supplement 1*). SNBP proteins in *D. melanogaster* tend to be short (<200 a.a.) and mostly have high isoelectric points (>10), consistent with their basic charge and potential function in tight packaging of DNA (*Table 1*). A closer examination revealed that 11 SNBP genes encode a single HMG box, whereas four genes (*Tpl94D, Prtl99C, Mst33A,* and *CG42355*) encode two HMG boxes (*Figure 1—figure supplement 2*).

To investigate the retention of SNBP genes across *Drosophila* species, we expanded our analysis to homologs of *D. melanogaster* SNBP genes found in published genome assemblies from 15 *Drosophila* species with NCBI gene annotation. We also included *Scaptodrosophila lebanonensis* as an outgroup species. Our phylogenomic analyses revealed that two SNBP genes (*tHMG* and *Prot*) underwent recent gene duplications in *D. melanogaster*. Both are present as closely related paralogs (*tHMG-1* and *tHMG-2, ProtA* and *ProtB*) in *D. melanogaster* but only in one copy in closely related species (*Figure 1*; *Tirmarche et al., 2014*; *Jayaramaiah Raja and Renkawitz-Pohl, 2005*). Five SNBP genes are found only in the *Sophophora* subgenus: *CG42355, Mst33A, Mst77F, Prtl99C,* and *Tpl94D* (*Figure 1*), and are, therefore, less than 40 million years old. At the other extreme, we found orthologs of eight *D. melanogaster* SNBP genes (*CG14835, CG30056, CG30356, CG31010, CG34529, ddbt, tHMG,* and *Prot*) in the outgroup species, *S. lebanonensis*. Thus, these eight SNBP genes are at least 50 million years old (*Suvorov et al., 2022*).

Our inability to detect homologs beyond the reported species does not appear to result from their rapid sequence evolution. Indeed, abSENSE analyses (*Weisman et al., 2020*) support the finding that *Prtl99C, Mst77F, Mst33A, Tpl94,* and *CG42355* were recently acquired in *Sophophora* within 40 MYA. For example, the probability of a true homolog being undetected for *Prtl99C* and *Mst77F* is 0.07 and 0.18 (using E-value = 1), respectively (*Supplementary file 1*, 'Materials and methods'). We also examined the syntenic regions of SNBP genes (conserved genomic neighboring genes) to confirm the loss of SNBP genes in some representative species, e.g., *D. kikkawai, D. ananassae, D. pseudoobscura, D. willistoni, D. albomicans, D. virilis,* and *S. lebanonensis*. Although abSENSE and synteny analyses rule out the absence of true homologs, they cannot rule out the less parsimonious possibility that SNBP genes are older but were lost multiple times in non-*Sophophora* species. Similarly, our analysis focuses on SNBP genes present in *D. melanogaster*, but other *Drosophila* species may have additional, unrelated SNBP genes.

**Table 1.** McDonald–Kreitman tests for positive selection on sperm nuclear basic protein (SNBP) genes in two *Drosophila* species.

| Name | Location (Mb) | Length | pI* | # of HMG | D. melanogaster Alpha | D. melanogaster $\chi^2$ p-value | D. serrata Alpha | D. serrata $\chi^2$ p-value | Expression stage† | Phenotype | Citations |
|---|---|---|---|---|---|---|---|---|---|---|---|
| CG30056 | 2R:12.6 | 137 | 11.05 (10.70) | 1 | -5 | 0.09 | 0.75 | 0.208 | Undefined | Undefined | a |
| CG30356 | 2R:8.7 | 149 | 10.65 (10.89) | 1 | 0.785 | **0.011** | 0.44 | **0.226‡** | Pre-individualization | Undefined | b,c |
| CG31010 | 3R:30.7 | 254 | 4.77 (8.10) | 1 | 0.535 | **0.034** | 0.61 | **0.001** | Undefined | Undefined | a |
| CG34269 | 3L:0.5 | 191 | 10.7 (10.34) | 1 | -0.263 | 0.613 | 0.736 | **0.001** | Undefined | Undefined | a |
| CG42355 | 3L:2.0 | 161 | 11.29 (10.89) | 2 | 0.682 | **0.012** | 0.682 | **0.036** | Undefined | Undefined | b,c |
| Mst33A | 2L:11.6 | 359 | 10.61 (10.14) | 2 | 0.208 | 0.391 | NA | NA | Undefined | Undefined | c |
| ddbt | 3L:0.3 | 117 | 12.3 (11.76) | 1 | -0.316 | 0.642 | -0.333 | 0.647 | Mature sperm | Sterile | d |
| Mst77F | 3L:20.8 | 215 | 10.34 (9.95) | 1 | -0.308 | 0.628 | NA | NA | Mature sperm | Sterile | e |
| Prtl99C | 2R:29.8 | 201 | 11.25 (10.57) | 2 | 0.242 | 0.483 | NA | NA | Mature sperm | Sterile | f |
| Tpl94D | 2R:23.0 | 164 | 11.3 (10.11) | 2 | 0.571 | **0.023** | 0.52 | **0.074‡** | Pre-individualization | Fertile | g |
| CG14835 | 3L:7.4 | 152 | 10.43 (4.81) | 1 | 0.332 | 0.416 | NA | NA | Mature sperm | Fertile | a |
| ProtA§ | 2R:14.9 | 146 | 11.12 (11.52) | 1 | 0.027 | 0.941 | 0.667 | **0.012** | Mature sperm | Low fertility | e |
| ProtB§ | 2R:14.9 | 144 | 10.8 (11.60) | 1 | -0.029 | 0.945 | 0.952 | **0** | Mature sperm | Low fertility | e |
| tHMG-1 | 3R:22.5 | 126 | 7.67 (6.11) | 1 | 0.659 | **0.02** | NA | NA | Pre-individualization | Fertile | b |
| tHMG-2 | 3R:22.5 | 133 | 8.94 (7.19) | 1 | 0.443 | 0.199 | NA | NA | Pre-individualization | Fertile | b |

We only show results from unpolarized MK tests using all (including rare) SNPs. Other variations of these results (e.g., polarized, excluding rare SNPs) are shown in *Supplementary file 5*.
Citations: [a]***Yamaki, 2018***, [b]***Gärtner et al., 2015***, [c]***Doyen et al., 2015***, [d]***Yamaki et al., 2016***, [e]***Jayaramaiah Raja and Renkawitz-Pohl, 2005***, [f]***Eren-Ghiani et al., 2015***, [g]***Rathke et al., 2007***.

Genes with any evidence of positive selection have p-values in bold.

*Isoelectric point of either the whole protein or just HMG domains only (in parentheses).

†Post-meiotic protein expression.

‡A significant signature of positive selection is obtained after removing low-frequency SNPs (<5%) and/or after polarizing changes (see *Supplementary file 5*).

§Independent duplications in two species.

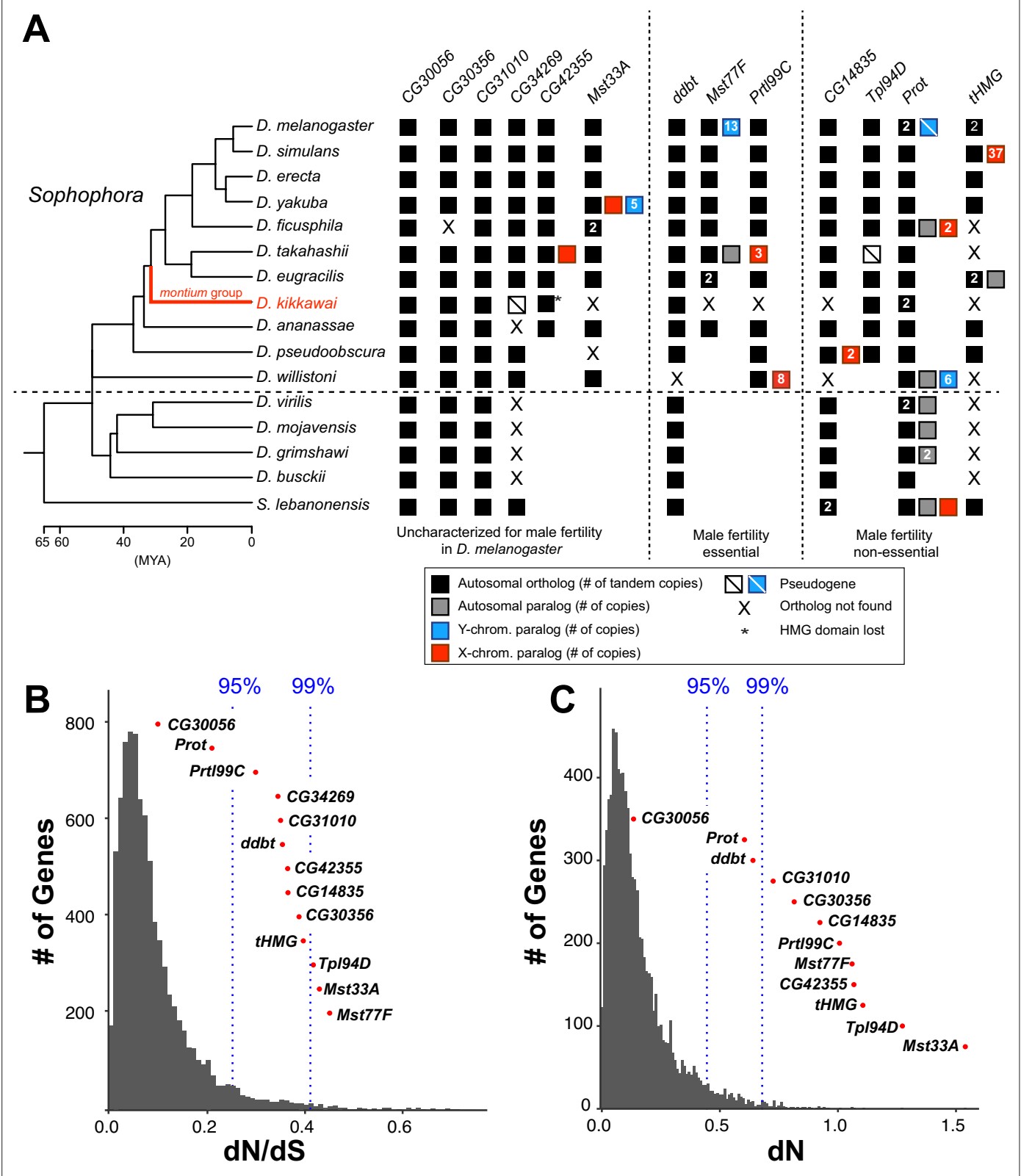

**Figure 1.** Origins and evolution of *Drosophila* sperm nuclear basic protein (SNBP) genes. (**A**) Phylogenomic analysis of 13–15 SNBP genes from *D. melanogaster* organized into three groups (dotted lines): required for male fertility, not required for male fertility, or untested in previous analyses. We identified homologs of these genes in 14 other *Drosophila* species and an outgroup species, *S. lebanonensis*, whose phylogenetic relationships and divergence times are indicated on the left (**Kumar et al., 2017**). Genes retained in autosomal syntenic locations are indicated by black squares, whereas

*Figure 1 continued on next page*

*Figure 1 continued*

paralogs located in non-syntenic autosomal locations, or X-chromosomes, or Y-chromosomes are indicated in gray, blue and red squares, respectively. Numbers within the squares show the copy number, if >1, of different genes, e.g., *D. melanogaster* has two paralogs each of both *Prot* and *tHMG* genes. An empty square with a line across it indicates that only a pseudogene can be found in the shared syntenic location, whereas an 'X' indicates that no ortholog is found, even though one is expected based on the phylogenomic inference of SNBP age. Based on this analysis, we infer that eight SNBP genes are at least 50 million years old, but only three genes are strictly retained in all 16 species (*CG30056*, *CG31010*, and *Prot*). Indeed, none of the SNBP genes required for male fertility in *D. melanogaster* are strictly conserved in other *Drosophila* species, either arising more recently (*Mst77F*, *Prtl99C*) or having been lost in at least one species after birth (*ddbt*). We also marked the *montium* group species, *D. kikkawai*, in red, because it has unusually lost six SNBP genes. (**B, C**) We compared dN/dS (**B**) or dN (**C**) values for all orthologous SNBP genes (red dots) in *D. melanogaster* compared to a histogram of the same values for the genome-wide distribution (gray bars) obtained from an analysis using six species by the 12 *Drosophila* genomes project (***Clark et al., 2007***). Our analyses reveal that most SNBP genes are at or beyond the 95th or 99th percentile for dN/dS or dN values (blue dashed lines). The values of *CG34269* are calculated using only five species because it is lost in one of the surveyed species, *D. ananassae*; therefore; we do not show its dN, as it is not comparable to other genes.

The online version of this article includes the following figure supplement(s) for figure 1:

**Figure supplement 1.** Expression patterns of sperm nuclear basic protein (SNBP) genes in *D. melanogaster spermatogenesis*.

**Figure supplement 2.** Number and location of high mobility group (HMG) boxes in sperm nuclear basic protein (SNBP) proteins.

**Figure supplement 3.** Expression patterns of sperm nuclear basic protein (SNBP) genes in *Drosophila* and *Scaptodrosophila* species.

**Figure supplement 4.** Sperm nuclear basic protein (SNBP) expression level in testes is correlated across *Drosophila* species.

We confirmed that *Drosophila* SNBP gene expression is primarily male-limited across species using publicly available RNA-seq data; their expression is particularly enriched in testes (***Figure 1—figure supplement 3A***; ***Supplementary file 2***). The only exception is a *CG42355* paralog in *D. takahashii* that also has weak expression in females (~9 TPM; ***Figure 1—figure supplement 3A***; ***Supplementary file 2***). We observed a moderate to high correlation (Spearman's rho = 0.142–0.753; ***Figure 1—figure supplement 4***) for the expression of SNBP genes between species. Like in *D. melanogaster*, most *Drosophila* SNBP proteins are small, possess at least one HMG box domain, and have high isoelectric points, suggesting that these features are crucial for their function (***Supplementary file 3***).

In addition to orthologs of these SNBP genes found in shared syntenic locations on autosomes, we also found sex chromosome-linked paralogs of SNBP genes in several species. The most dramatic example is the presence of 34 copies of *tHMG* paralogs in the poorly assembled X chromosomal region of *D. simulans* (***Figure 1***). These are discussed in more detail later in this study.

## Rapid evolution and positive selection of *Drosophila* SNBP genes

Based on the precedent of rapidly evolving protamines in mammals, we next investigated whether *Drosophila* SNBP genes also evolve rapidly. We calculated protein evolution rates (non-synonymous substitution rates over synonymous substitution rates, dN/dS) for 13 of 15 *D. melanogaster* SNBP genes for six species in the *melanogaster* group (***Supplementary file 4***). We excluded two SNBP genes, *ProtB* and *tHMG-2,* since these duplicates are not found outside *D. melanogaster*. We found that 11 of 13 SNBP genes (except *CG30056* and *ProtA*) evolve faster (higher dN/dS) than 95% of protein-coding genes across the genome (***Figure 1B***). These high protein evolution rates are due to high dN instead of low dS (***Figure 1C***), suggesting that SNBPs evolve under either extensive positive selection or reduced functional constraints.

We used McDonald–Kreitman tests to test the possibility of recent positive selection in the *D. melanogaster* lineage, taking advantage of many sequenced strains from this species (***Rathke et al., 2007***; ***Witt et al., 2021***; ***Yang, 2007***; ***Yang et al., 2000***; ***Kasinathan et al., 2020***; ***Bayes and Malik, 2009***; ***Kim et al., 2021***; ***Altschul et al., 1990***; ***Manni et al., 2021***; ***Vedanayagam et al., 2022***). The McDonald–Kreitman test compares the ratio of non-synonymous to synonymous substitutions fixed during inter-species divergence to the ratio of these polymorphisms segregating within species. If there is an excess of non-synonymous changes fixed during inter-species divergence, this results from positive selection. Indeed, our tests reveal that five SNBP genes in *D. melanogaster* have evolved under positive selection during its divergence from *D. simulans* (***Table 1***). By polarizing the test using the inferred ancestral sequences of *tHMG-1* and *tHMG-2*, we find that *tHMG-1*, but not its paralog, *tHMG-2*, evolved under positive selection in *D. melanogaster* (***Supplementary file 5***).

We also took advantage of genome sequences from 110 *D. serrata* strains to carry out McDonald–Kreitman tests of SNBP genes from *D. serrata* compared to its sister species, *D. bunnanda* in the

*montium* group (**Reddiex et al., 2018**; **Bronski et al., 2020**). Among the SNBP genes shared between *D. melanogaster* and *D. serrata*, we found that four genes (*CG30356*, *CG31010*, *CG42355*, *Tpl94D*) evolved under positive selection in both *D. melanogaster* and *D. serrata*, whereas two genes (*CG30056*, *ddbt*) do not show a signature of positive selection in either species (**Supplementary file 5**). Three additional genes (*CG34269* and two *ProtA/B* duplicates) evolved under positive selection only in *D. serrata* (*ProtA/B* underwent independent gene duplications in *D. melanogaster* and *D. serrata*).

Finally, we used maximum-likelihood analyses using the site model of the codeml program in the PAML package (**Yang, 2007**; **Yang et al., 2000**) to investigate whether any residues in the SNBP genes had evolved under recurrent positive selection. We limited our analyses to 17 species of the *melanogaster* group to avoid saturation of synonymous substitutions. Although we recapitulated a previously published positive selection result using *ddbt* genes from just five *Drosophila* species (**Yamaki et al., 2016**), analyses using 17 *melanogaster* group species did not find a significant signature of site-specific positive selection in any SNBP gene (**Supplementary file 6**). The discrepancy between the McDonald–Kreitman tests and the PAML results indicates that although many SNBP genes evolve under positive selection, either SNBP genes or the exact residues evolving under recurrent positive selection vary throughout *Drosophila* evolution.

## What determines SNBP essentiality for male fertility

Of 15 SNBP genes, nine have been previously characterized for their roles in male fertility based on gene knockout or knockdown experiments in *D. melanogaster* (**Table 1**). These genes show differences in their importance for male fertility in *D. melanogaster*. Three genes (*Mst77F*, *Prtl99C*, and *ddbt*) are essential for male fertility (**Tirmarche et al., 2014**; **Kimura and Loppin, 2016**; **Jayaramaiah Raja and Renkawitz-Pohl, 2005**; **Eren-Ghiani et al., 2015**; **Rathke et al., 2010**), but separate knockouts of six individual genes (*CG14835*, *ProtA*, *ProtB Tpl94D*, *tHMG-1*, and *tHMG-2*) do not appear to impair male fertility under standard laboratory conditions. Further experimentation has revealed a fertility cost for double knockouts of *ProtA/ProtB* but only in conditions of sperm exhaustion via mating with excess numbers of females (**Tirmarche et al., 2014**). No information is currently available for the remaining six SNBP genes. We also found nearly strict retention of all SNBP genes in all sequenced strains of *D. melanogaster*, no matter whether they are essential for male fertility in laboratory experiments or not (**Supplementary file 7**).

There are a few distinguishing characteristics common to SNBP genes required for male fertility. Neither the number of HMG domains nor expression levels of SNBP are associated with essentiality. Instead, proteins essential for male fertility (*Mst77F*, *Prtl99C*, and *ddbt*) are more likely to be present in the mature sperm head, whereas transition SNBPs (*Tpl94D*, *tHMG-1*, and *tHMG-2*) are more likely to be dispensable, potentially due to functional redundancy with each other. Moreover, SNBP genes important for male fertility in *D. melanogaster* show no signature of positive selection according to McDonald–Kreitman tests. In contrast, two of the three identified transition SNBP genes evolve under positive selection (*Tpl94D* and *tHMG-1*). This suggests that genes with redundant function or less critical for male fertility are more likely to evolve under positive selection, although we note that several SNBP genes remain functionally uncharacterized or have not been tested exclusively (**Table 1**).

How does SNBP essentiality in *D. melanogaster* correlate with age and retention across *Drosophila* species evolution? We find that two essential SNBP genes (*Prtl99C* and *Mst77F*) are evolutionarily young, i.e., they arose relatively recently in *Drosophila* evolution. Moreover, both genes have been lost at least once in the *montium* group species since their birth. The third essential SNBP gene, *ddbt*, arose before the separation of *Drosophila* and *S. lebanonensis*, but it has also been lost at least once (in *D. willistoni*) among the 15 species analyzed (**Yamaki et al., 2016 Figure 1A**). Based on these findings, we infer that not only are these three essential SNBP genes subject to evolutionary turnover, but they also gain or lose essential function across *Drosophila* evolution. Our findings are reminiscent of recent studies that show the high evolutionary lability of many genes involved in essential heterochromatin or centromere function in *Drosophila* (**Kasinathan et al., 2020**; **Bayes and Malik, 2009**).

## *CG30056* is dispensable for male fertility in *D. melanogaster* despite being universally retained in *Drosophila* species

Given the high evolutionary turnover of SNBP genes in our sampling of 15 *Drosophila* species, we investigated whether any SNBP genes are universally retained across all *Drosophila* species. For this

purpose, we expanded our phylogenomic analysis of SNBP evolution to a recently published dataset of 78 highly contiguous and complete *Drosophila* genomes (*Kim et al., 2021*), using tblastn and reciprocal blastx searches (*Altschul et al., 1990*). Based on this analysis, we find only two SNBP genes have been strictly retained across all *Drosophila* species — *Prot* and *CG30056* (*Figure 1A*), while the *CG31010* gene has only been lost in one species. *Prot* and *CG30056* also have the lowest dN/dS among SNBP genes (*Figure 1B*), suggesting that they evolve under a higher degree of selective constraint than other SNBP genes.

Previous studies have shown that a dual loss of *ProtA/ProtB* reduces male fertility in conditions of sperm exhaustion in *D. melanogaster* (*Tirmarche et al., 2014*). However, *CG30056* remains functionally uncharacterized for its role in male fertility despite being one of only two well-retained and highly conserved SNBP genes in *Drosophila* species (*Figure 1B*). To study its contribution to male fertility, we generated a complete deletion knockout of *CG30056* gene using CRISPR/Cas9 (*Figure 2A*). *CG30056* is located in an intron of *frazzled*, a gene essential for development and morphogenesis (*Figure 2A*). We co-injected a construct encoding two gRNAs designed to target sequences immediately flanking *CG30056* together with a repair construct containing 3xP3-DsRed (a visible eye marker). We successfully obtained transgenic flies encoding DsRed and validated the deletion of *CG30056* using PCR (*Figure 2A*). Homozygous knockout flies were viable, confirming that the removal of *CG30056* did not disrupt the essential *frazzled* gene.

Next, we tested how *CG30056* contributes to male fertility. We first generated *CG30056-KO* homozygous males by crossing two different *CG30056-KO* founder lines either to each other or to a *D. melanogaster* strain with a large deletion spanning *CG30056* (*Df(2R)BSC880*). This 'trans-heterozygote' strategy prevents our phenotypes from being affected by CRISPR/Cas9-generated off-target mutations generated in the same founder chromosome. As controls, we used *CG30056/SM6a* or *CG30056/CyO* heterozygous males (*SM6a* and *CyO* are balancer chromosomes with an intact copy of *CG30056)*. We compared the fertility of these males by mating them with two wildtype females at room temperature and counting their adult offspring. We found only modest differences in offspring number between heterozygous controls and homozygous knockout males (*Figure 2B*; *Supplementary file 8*), implying that loss of *CG30056* does not significantly lower male fertility. We also detected no evidence of sex-ratio distortion in our crosses (*Figure 2C*; *Supplementary file 8*). Parallel fertility experiments conducted with mutants of *Mst77F* or *Tpl94D* recapitulated previous findings, confirming the essentiality of *Mst77F* and the dispensability of *Tpl94D* (*Figure 2B*; *Supplementary file 8*; *Gärtner et al., 2015*; *Kimura and Loppin, 2016*).

Comparisons to balancer chromosome-containing males could mask more subtle fertility effects of *CG30056-KO*. Therefore, we increased the stringency of our fertility assay in two ways. First, we compared the fertility of *CG30056-KO* 'trans-heterozygote' males with heterozygous males carrying one *CG30056-KO* allele and one wildtype chromosome, which was the parental chromosome used to create the deletion. Second, we carried out fertility experiments by crossing individual males with 10 wildtype females to exhaust male sperm; similar assays were used to reveal fertility consequences of *ProtA/ProtB* double deletions (*Tirmarche et al., 2014*). In spite of this increased stringency, we did not find a significant fertility difference between *CG30056-KO* and the control sibling males (*Figure 2B*). Thus, despite its strict retention for more than 50 million years of *Drosophila* evolution, standard laboratory conditions reveal no male fertility requirement of *CG30056*. However, *CG30056* might play another function that we did not assay for, such as in sperm storage or sperm precedence that would explain its strict retention.

Thus, two of three SNBP genes (*Mst77F* and *Prtl99C*) essential for male fertility are evolutionarily young and poorly retained. In contrast, one of the most well-conserved and well-retained SNBP genes (*CG30056*) is not essential. Our work suggests that newly arisen *Drosophila* SNBP genes are more likely to encode essential, non-redundant male fertility functions than ancient, well-retained ones.

## Recurrent amplification of a subset of SNBP genes on sex chromosomes

Our analysis of SNBP genes across a limited set of *Drosophila* species had already revealed significant evidence of evolutionary turnover (*Figure 1*). We further analyzed evolutionary turnover of SNBP genes in 78 *Drosophila* species and two other *Drosophilidae* species, most of which lack detailed gene annotation (*Kim et al., 2021*). We inferred gains and losses of SNBP genes in these species, which we

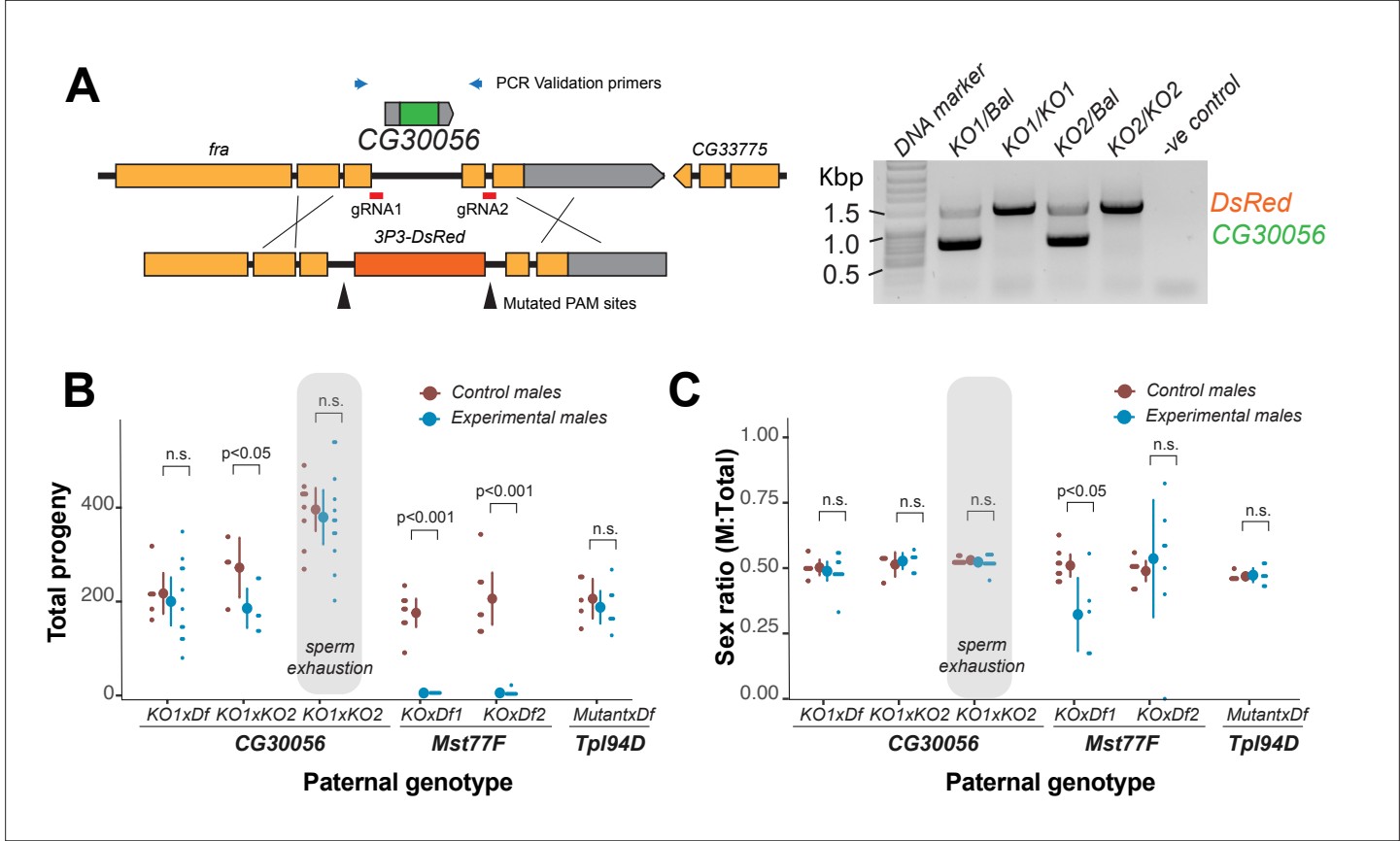

**Figure 2.** The strictly retained, highly conserved sperm nuclear basic protein (SNBP) gene, *CG30056*, is dispensable for male fertility in *D. melanogaster*. (**A**) The SNBP gene, *CG30056*, is encoded co-directionally in an intron of the essential *frazzled* gene. Using guide RNAs designed to match sites flanking *CG30056*, and a healing construct encoding eye-specific DsRed, we created a knockout allele replacing *CG30056* with *DsRed*. The knockout was verified using PCR and primers flanking the *CG30056* locus (right). Note that balancer lines encode a wildtype copy of *CG30056*. (**B**) We performed fertility assays comparing *CG30056* homozygous knockout flies with heterozygous controls, either KO/Balancer or KO/wt (gray ovals). Each dot represents a single replicate, and the average and 95% confidence interval based on standard errors are shown in the figures. Fertility assays were performed either for a few days or to sperm exhaustion (gray ovals). We also assayed fertility of knockout strains for the fertility-essential *Mst77F* gene, and the fertility-nonessential *Tpl94D* gene. We also documented the sex ratios of the resulting progeny in (**C**). Consistent with previous findings, we found that *Mst77F* knockout males are essentially sterile and *Tpl94D* knockout males were indistinguishable from their heterozygous controls. We found either no or weak evidence of fertility impairments in two different crosses with homozygous *CG30056* knockout males compared to KO/Balancer controls. However, we found no evidence of *CG30056* requirement for male fertility in more stringent 'sperm exhaustion' fertility experiments compared to KO/wildtype controls (gray ovals). (**C**) We observed no significant evidence of sex-ratio distortion that would suggest an X-versus-Y meiotic drive in progeny resulting from either *CG30056*, *Mst77F*, or *Tpl94D* knockout males. Although there is suggestive evidence of sex-ratio distortion in progeny of one of the *Mst77F* genotypes, this is inconsistent between the two crosses and most likely due to stochastic effects of having very few resulting progeny. The raw data of (**B**) and (**C**) are shown in *Supplementary file 8*.

The online version of this article includes the following source data for figure 2:

**Source data 1.** Uncropped gel image corresponding to *Figure 2C*.

**Source data 2.** Raw gel image corresponding to *Figure 2C*.

represent on a circular phylogram of all species. To assign the chromosomal location of SNBP genes, we estimated coverage of publicly available Illumina and Nanopore reads (represented in *Figure 3* and *Supplementary file 9*) of either male or mixed-sex flies from various *Drosophila* species. We also assigned location to specific Muller elements based on 3285 BUSCO (Benchmarking Universal Single-Copy Orthologs) genes on the contigs (*Manni et al., 2021*). We could readily assign male-specific regions to the Y chromosomes for species where sequencing reads were available from male and female flies separately. However, we could not ascribe a sex-chromosomal linked location of a contig to either the X or Y chromosome in cases where there was no linkage information from BUSCO genes and no read data available from females, only from males and mixed-sex flies.

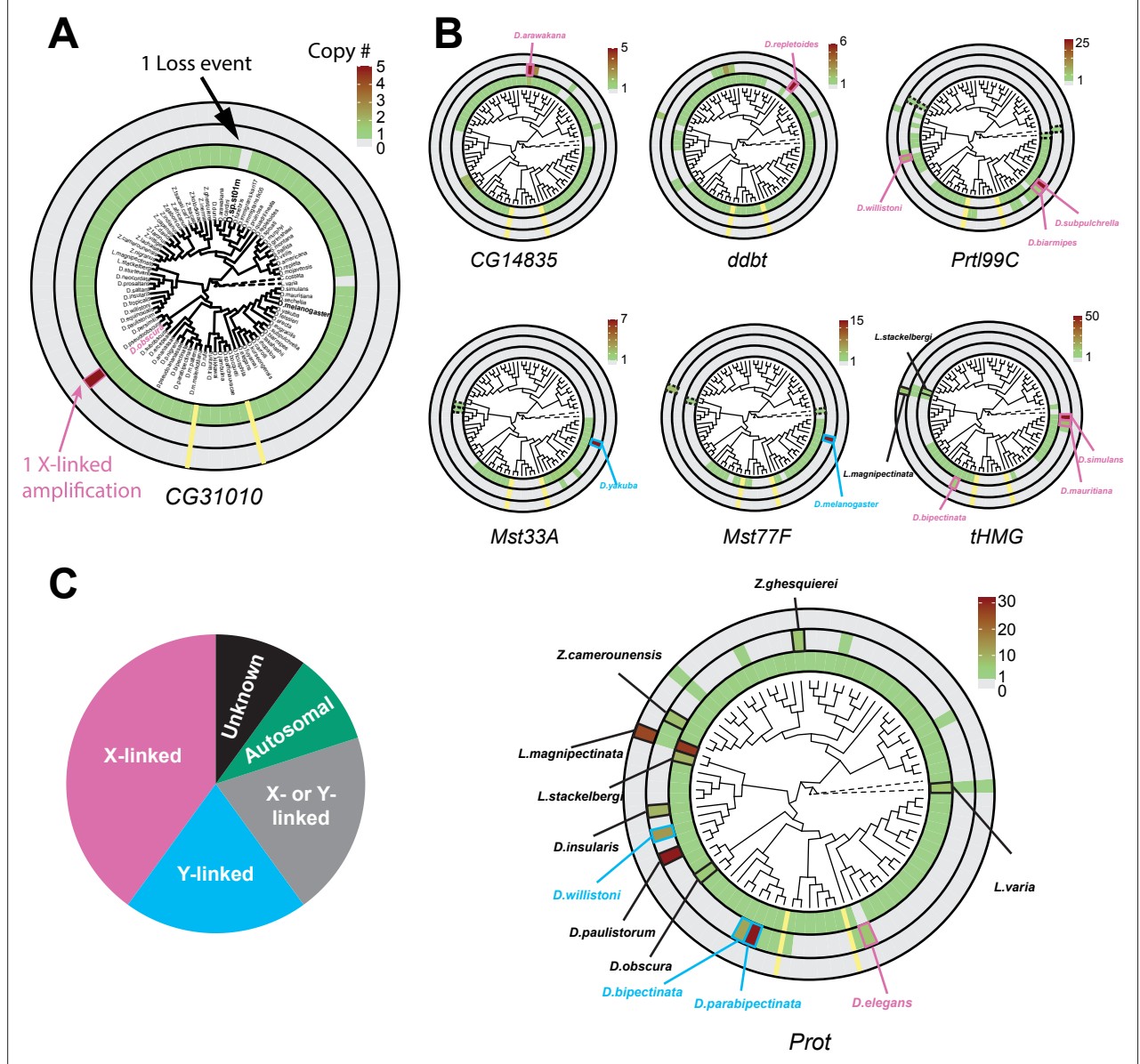

**Figure 3.** Recurrent amplifications of *Drosophila* sperm nuclear basic protein (SNBP) genes are biased for sex-chromosomal linkage. (**A**) Using reciprocal BLAST (see 'Materials and methods'), we searched for homologs of each *D. melanogaster* SNBP gene in 78 distinct *Drosophila* species and two outgroup species (shown in dot lines). We depict our findings using the circular phylogram representation for SNBP gene *CG31010*. The innermost circle is a circular phylogeny of the species (*Kim et al., 2021*). The next circle ring indicates autosomal copies, with colors to indicate copy number (scale bar, top left; note that scales are different for each gene). Thus, *CG31010* is present in one autosomal copy in all but one *Drosophila* species (gray bar). The third circle indicates sex-chromosomal copies. Red and blue frames in the middle ring indicate X- or Y-linkage if that can be reliably assigned. Dotted frames indicate copies that might not be real orthologs based on phylogeny, whereas solid frames indicate five or more copies. For example, *CG31010* is present in five copies on the X-chromosome of *D. obscura*. The outermost circle shows copies with ambiguous chromosomal location: there are no such copies for *CG31010*. (**B**) Using the same representation scheme, we indicate gene retention and amplification for seven other SNBP genes for which we find robust evidence of amplification, from a copy number of five (*CG14835*) to nearly 50 (*tHMG*). We also marked the *montium* group species that lost many SNBP genes with yellow lines. We note that assemblies of *Lordiphosa* species have lower quality, and the data need to be interpreted carefully. (**C**) SNBP gene amplifications (five or more copies) are heavily biased for sex chromosomal linkage. Given the relative size of sex chromosomes and autosomes, this pattern is highly non-random (test of proportions, p=2.3e-5).

The online version of this article includes the following figure supplement(s) for figure 3:

**Figure supplement 1.** Six sperm nuclear basic protein (SNBP) genes did not undergo significant gene amplification events in *Drosophila* species.

**Figure supplement 2.** Concerted evolution of sperm nuclear basic protein (SNBP) gene amplifications.

All ancestrally-retained SNBP genes are located on autosomes in *D. melanogaster*. This is also the case in most *Drosophila* species. On the circular phylogram, we represented losses or duplications of the ancestral autosomal gene in the innermost circle, gains on the sex chromosomes are represented in the next (middle) concentric circle, and gains with ambiguous chromosomal location are represented in the outermost concentric circle (*Figure 3*, *Figure 3—figure supplement 1*). We use the *CG31010* gene to illustrate this representation. *CG31010* has been retained in all *Drosophila* species except one, which is shown by a gray bar in the innermost concentric circle. In addition, *CG31010* amplified to a total of five X-linked copies in *D. obscura*, represented by a dark red bar in the middle concentric circle (*Figure 3A*).

Our expanded survey reinforced our initial findings (*Figure 1A*) that multiple SNBP genes are subject to lineage-specific amplifications (more than five copies in one species). We found that 8 of 13 *D. melanogaster* SNBP genes investigated (*CG14835, ddbt, Mst33A, Mst77F, Prtl99C, Prot, tHMG,* and *CG31010)* underwent amplification in at least one species (*Figure 3* and *Table 2*). In total, we found that SNBP genes have experienced 20 independent amplification events, including one event in the outgroup species, *Leucophenga varia* (*Figure 3A and B*). Most SNBP amplifications are evolutionarily young (<10 million years old; *Figure 3—figure supplement 2*), and 15 of them are specific to a single surveyed species (*Figure 3A and B*). Like their parental genes, most amplified copies encode positively charged HMG domain-containing proteins (*Supplementary file 3*) and have male-specific, mostly testis-specific, expression (*Figure 1—figure supplement 3B*). Thus, amplified copies are also likely to function as SNBP genes. Many amplified SNBP genes are arranged in tandem arrays, making their sequences hard to assemble. Moreover, some amplified SNBP genes, e.g., *Dox*-related genes derived from *ProtA/B* in *D. simulans* (*Vedanayagam et al., 2021*; *Muirhead and Presgraves, 2021*), are too diverged from the parental genes for unambiguous assignment and are missing in our survey. Therefore, our reported numbers are thus likely underestimates.

We found that eight amplifications are X-linked and four are Y-linked, whereas, for four amplifications, we can infer sex chromosomal linkage but cannot distinguish between X- or Y-linkage (*Figure 3*; *Supplementary file 10*). Thus, we conclude that 80% (16/20, *Figure 3B*) of amplifications occurred on sex chromosomes. This high fraction is significantly higher than the null expectation (Test of Proportions, p=2.3e-5) if SNBP amplifications were randomly distributed between sex chromosomes or autosomes. Under the null expectation, ~33% should be on sex chromosomes, given each chromosome arm, except the dot chromosome, has a similar size.

To better understand the evolutionary origins and potential function of SNBP gene amplifications, we investigated the amplification of *tHMG* in *D. simulans* and sister species (*Figure 4*). We traced its evolutionary history using a combination of phylogenetic and shared synteny analyses. From these analyses, we inferred that *tHMG* copies on the heterochromatic X chromosome (*tHMG-hetX*) experienced two duplications before the amplification. The first of these occurred onto the euchromatic X region, proximal to *CG12691*. Subsequently, a second duplication spanning X-linked *tHMG* and part of *CG12691* occurred onto pericentromeric X (*Figure 4A*). We analyzed their sequences to assess the evolutionary pressures shaping the amplified *tHMG-hetX* relative to the parental *tHMG* copies (*tHMG-Anc*). Within an 81 amino acid residue stretch, we found 19 non-synonymous changes that occurred within *tHMG-hetX* compared to just 4 non-synonymous changes that occurred within *tHMG-Anc* (*Figure 4B*). Thus, *tHMG-hetX* evolves considerably faster than *tHMG-Anc.* Branch analyses using PAML revealed that both *tHMG-hetX* and *tHMG-Anc* branches have significantly higher protein evolution rates than branches in other species (*Figure 4C*; $\omega$ = 1.6, LRT test, p=0.007; *Supplementary file 11*). However, the branch-site test did not reveal clear evidence of positive selection on any residues (LRT test, p=0.23; *Supplementary file 11*). Our analyses on the *tHMG* expansion thus reveal a rapid, complex series of chromosome rearrangements leading to sex chromosome SNBP amplifications, which bear hallmarks of positive selection.

## Short life of sex chromosome-linked SNBP genes

Previous studies showed that the *Dox* meiotic driver arose from an SNBP partial gene amplification (*ProtA/B*) in *D. simulans* (*Vedanayagam et al., 2021*; *Muirhead and Presgraves, 2021*; *Vedanayagam et al., 2022*). We hypothesize that the *Drosophila* sex-chromosomal SNBP amplifications we have found might similarly be involved in genetic conflicts between sex chromosomes across the *Drosophila* genus, via X-versus-Y meiotic drive. In contrast, all ancestral single-copy SNBP genes that

**Table 2.** Summary of evolution events of *Drosophila* sperm nuclear basic protein (SNBP) genes.

| Name | Phenotype | Age (My) | Expression level (TPM)* | Evolutionary rate (dN/dS)† | Positive selection (MK test)‡ | Amplification event(s)§ | Number of loss events in 80 species¶ | Number of loss events in the montium group (X-Y fusion) |
|---|---|---|---|---|---|---|---|---|
| CG30356 | Undefined | >65 | 353 | 0.39 | + | 0 | 1 | 0 |
| CG31010 | Undefined | >65 | 346 | 0.35 | + | 1X | 1 | 0 |
| CG34269 | Undefined | >65 | 49 | 0.34 | + | 0 | 2 | 5 |
| CG42355 | Undefined | 35 | 532 | 0.36 | + | 0 | 0 | 0 |
| Mst33A | Undefined | 35 | 632 | 0.43 | - | 1Y | 1 | 1 |
| ddbt | Sterile | >65 | 130 | 0.35 | - | 1X | 3 | 0 |
| Mst77F | Sterile | 35 | 3636 | 0.45 | - | 1Y | 3 | 1 |
| Prtl99C | Sterile | 45 | 540 | 0.30 | - | 2X | 4 | 2 |
| CG30056 | Fertile | >65 | 406 | 0.10 | - | 0 | 0 | 0 |
| CG14835 | Fertile | >65 | 432 | 0.36 | - | 1X | 3 | 1 |
| Tpl94D | Fertile | 45 | 564 | 0.41 | + | 0 | 2 | 0 |
| Prot | Fertile | >65 | 1118,1619 | 0.21 | + | 2A;1X;2Y;4X/Y;1U | 0 | 0 |
| tHMG | Fertile | >65 | 935,1194 | 0.39 | + | 2X; 1U | 5 | 1 |

\* Gene expression level in *D. melanogaster* testes (**Supplementary file 2**).

†dN/dS in *D. melanogaster* subgroup species (**Supplementary file 4**).

‡Positive selection based on McDonald–Kreitman tests in *D. melanogaster* and/or *D. serrata*.

§Any specific location with five or more copies of any one SNBP gene. A represents all autosomes combined, X represents the X chromosome, Y represents the Y chromosome, X/Y represents either the X or Y chromosome, and U represents regions with unknown chromosome locations.

¶Number of potential loss events inferred by the phylogeny (**Figure 3** and **Figure 3—figure supplement 1**). Some of these may represent false negatives due to incomplete genome assemblies.

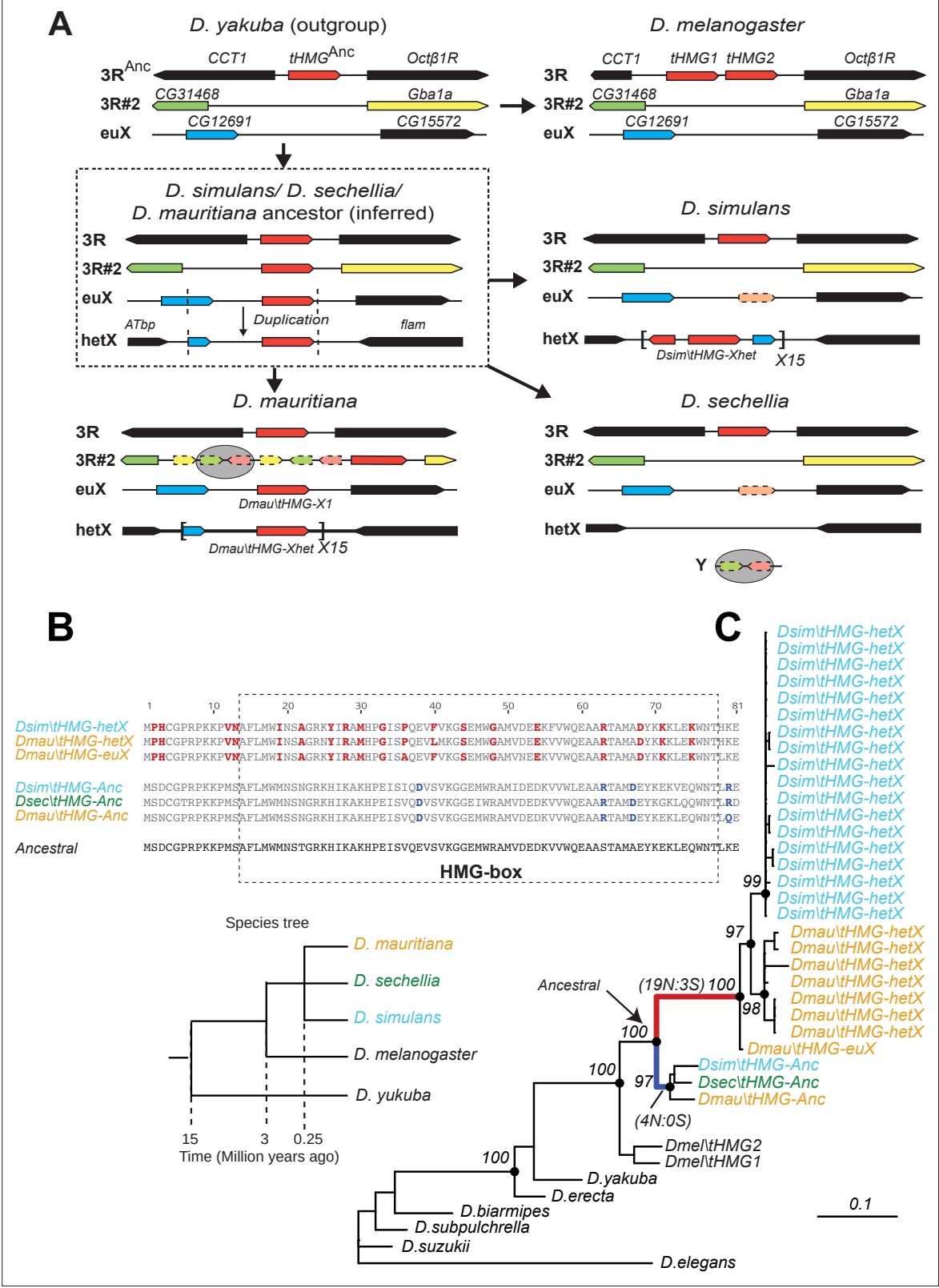

**Figure 4.** Tracing the duplication and amplification of *tHMG* genes in *D.simulans* and close relatives. (**A**) Using a combination of genome assemblies and phylogenetic analyses, we traced the evolutionary origins and steps that led to the massive amplification of *tHMG* genes on the *D. simulans* X chromosome. The first step in this process was the duplication of the ancestral *tHMG* gene (flanked by *CCT1* and *Octb1R*) on the 3R chromosomal arm to a new location on 3R (*tHMG-3R#2* now flanked by *CG31468* and *Gba1a*) and to a location on the X chromosome euchromatin, where *tHMG-euX*

*Figure 4 continued*

is flanked by *CG12691* and *CG15572*. We infer that this *CG12691-tHMG-euX* locus then duplicated to another locus in X-heterochromatin, between *Atbp* and the *flamenco* locus, and further amplified. These resulting copies experienced different fates in *D. simulans* and its sibling species. For example, in *D. sechellia*, *tHMG-3R#2*, *tHMG-euX*, and *tHMG-hetX* were all lost but a degenerated copy of *tHMG-3R#2* and flanking genes can be found on its Y chromosome. In contrast, in *D. mauritiana*, *tHMG-3R#2* pseudogenized on 3R, *tHMG-euX* was retained while *tHMG-hetX* underwent an amplification to a copy number of 15 tandemly arrayed genes in the X heterochromatin. Finally, in *D. simulans*, *tHMG-3R#2* was completely lost, *tHMG-euX* was pseudogenized, and *tHMG-hetX* amplified to a copy number of 15 on the X heterochromatin. We note that the amplification unit sizes are different between *D. simulans* and *D. mauritiana*, suggesting that these were independent amplifications. Moreover, we detected different copy numbers (all more than 30) of *tHMG-hetX* across three sequenced strains of *D. simulans* we surveyed. This difference is likely due to both incomplete assemblies of this region and strain-specific differences. In addition to this X chromosomal expansion, we also found a few degenerated copies of *tHMG* on the 3R heterochromatic region and the Y chromosome. (**B**) The alignment shows the divergence between different *tHMG* copies in the *D. simulans* clade and *D. melanogaster*. Surprisingly, we X-linked *tHMG* duplicates diverged more from parental genes on autosomes, indicating that they experienced different evolutionary forces than the parental copies. Among 243 aligned nucleotide sites, we found 19 non-synonymous changes and only 3 synonymous changes shared in all X-linked copies after they diverged from the parental copy. Similarly, four non-synonymous changes and no synonymous change occurred on the parental copy in the ancestral species of the *simulans* clade. Most non-synonymous changes are in the DNA-binding HMG box. As a result, parental copies and new X-linked copies in *D. simulans* and *D. mauritiana* only share ~70% protein identity, which is very low given the <3 MY divergence. Our branch test using PAML further shows that both branches have significantly higher protein evolution rates ($\omega$ = 1.6, LRT test, p=0.007; *Supplementary file 11*). However, we did not find evidence of positive selection using a branch-site test (LRT test, p=0.23; *Supplementary file 11*). (**C**) Phylogenetic analyses of the various *tHMG* genes confirm the chronology of events outlined in (**A**) and find strong evidence of concerted evolution among the amplified *tHMG-hetX* copies on *D. mauritiana* and *D. simulans*, in which copies from the X-linked heterochromatic region are highly homogeneous within species, but diverged between species. For comparison, we showed the species tree on the left, and the phylogeny of three *D. simulans* clade species is not solved due to lineage sorting and gene flow. To simplify the analysis, we only used sequences that are annotated in NCBI databases.

gave rise to these amplifications are encoded on autosomal loci and thus are more likely to encode suppressors of meiotic drive, as is the case for the *ProtA/B* genes against *Segregation Distorter* (*Gingell and McLean, 2020*).

If our hypothesis for this duality of SNBP gene functions is correct, we would further predict that ancestrally autosomal SNBP genes that became linked to sex chromosomes would preferentially amplify to become meiotic drivers or be lost due to the loss of ancestral functional requirements. To test this hypothesis, we surveyed SNBP genes that became linked to sex chromosomes via chromosome fusions. We found three SNBP genes, *CG14835*, *CG34269*, and *ddbt*, which are widely retained in *Drosophila* species on the ancestrally autosomal Muller D element. Both *CG14835* and *ddbt* genes independently became X-linked in the *D. willistoni*, *D. pseudoobscura*, and *D. repletoides* clades, while *CG34269* became X-linked in the two former clades. Consistent with our hypothesis, some newly X-linked SNBP genes either degenerated (two instances out of eight) or translocated back to autosomes (one instance out of eight). In five cases, SNBP genes were still retained on sex chromosomes (*Figure 5*; *Supplementary file 12*). Among these five cases, we observed one amplification event –*ddbt* amplified to six copies in *D. repletoides* (*Figure 3B*), consistent with the idea that it may act as a meiotic driver. In contrast, we found no instances of pseudogenization or subsequent translocation to the X chromosome of SNBP genes that are still preserved on their original autosomal locations or involved in chromosome fusions between autosomes (0/16). This difference is highly significant (*Figure 5*; *Supplementary file 12*; 3:5 versus 0:16, Fisher's exact test, p=0.03). However, this pattern may have alternate explanations, including previous findings that male-biased genes preferentially avoid the X chromosome (*Sturgill et al., 2007*).

## Loss of SNBP genes in the *montium* group coincides with X-Y chromosomal fusion

Our phylogenomic analyses also highlighted one *Drosophila* clade—the *montium* group of species (including *D. kikkawai*)—which suffered a precipitous loss of at least five SNBP genes (*Figure 3*). In contrast to *D. kikkawai*, three of these five SNBP genes are retained in all other *D. melanogaster* group species. For the other two genes, *Mst77F* was lost twice and *tHMG* was lost three times (*D. fuyamai* has lost both *Mst77F* and *tHMG*). A closer examination allowed us to infer that six different SNBP genes underwent 11 independent degeneration events in the *montium* group (*Figure 6A*). Intriguingly, five of six SNBP genes lost in the *montium* clade (*Mst77F*, *Prtl99C*, *Mst33A*, *tHMG*, *CG14835*)

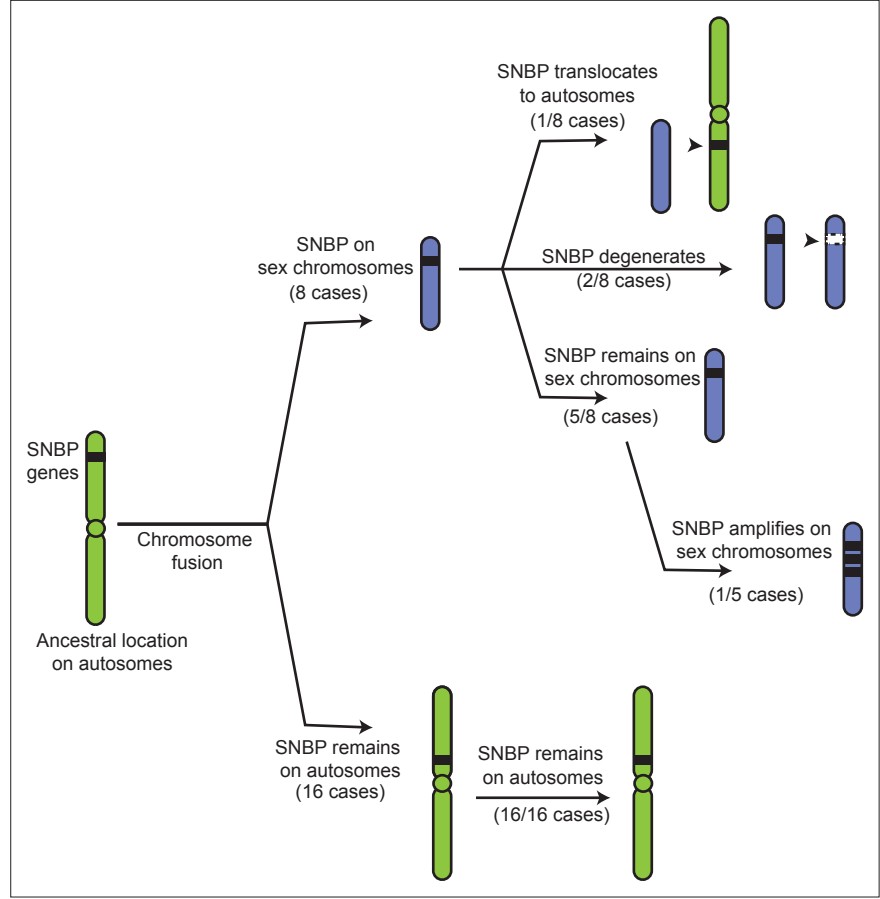

**Figure 5.** Evolutionary retention, degeneration, or translocation of sperm nuclear basic protein (SNBP) genes following chromosomal fusions. SNBP genes are ancestrally encoded on autosomes. Following chromosome fusion over *Drosophila* evolution, we found eight cases in which three SNBP genes (*CG14835, CG34269,* and *ddbt*) became linked to sex chromosomes. In 1/8 cases, SNBP genes translocated back to an autosome. In 2/8 cases, the sex chromosome-linked SNBP genes degenerated despite being otherwise widely conserved in non-*montium Drosophila* species. In 5/8 cases, SNBP genes were retained on neo-sex chromosomes in 5/8 cases. Among these, we observed one amplification event; *ddbt* amplified to six copies in *D. repletoides*. In contrast to sex chromosomal linkage, SNBP genes that remained linked to autosomes despite chromosomal fusions were strictly retained in 16/16 cases. These retention patterns differ significantly between sex chromosomes and autosomes (Fisher's exact test, p=0.03).

are also among the eight SNBP genes subject to sex chromosome-specific amplifications in other *Drosophila* species (**Figure 3** and **Table 2**).

Notably, we did not find SNBP amplification events in any species of the *montium* clade. Given our hypothesis that autosomal SNBP genes might be linked to the suppression of meiotic drive (above), we speculated that the loss of these genes in the *montium* group of *Drosophila* species may have coincided with reduced genetic conflicts between sex chromosomes in this clade.

How could such reduction in sex chromosomal genetic conflicts arise? An important clue came from a previous study, which showed that many ancestrally Y-linked genes are present in females because of possible relocation to other chromosomes in the *montium* group (**Dupim et al., 2018**). We revisited this question to pinpoint which Y chromosomal gene translocations coincided with SNBP degeneration in this lineage. Using the available assemblies with Illumina-based chromosome assignment, we surprisingly found that most ancestrally Y-linked genes are actually linked to the X chromosome (**Figure 6A**). For example, we were able to unambiguously infer X-chromosomal linkage for most ancestrally Y-linked genes in *D. kikkawai* (7/10), *D. jambulina* (9/11), *D. bocqueti* (7/10), *D. aff. chauvacae* (7/8), and *D. triauraria* (11/12) (**Figure 6A**), although some ancestrally Y-linked genes are not present in the assemblies. Moreover, in *D. triauraria*, we found that 11 of 12 ancestrally Y-linked

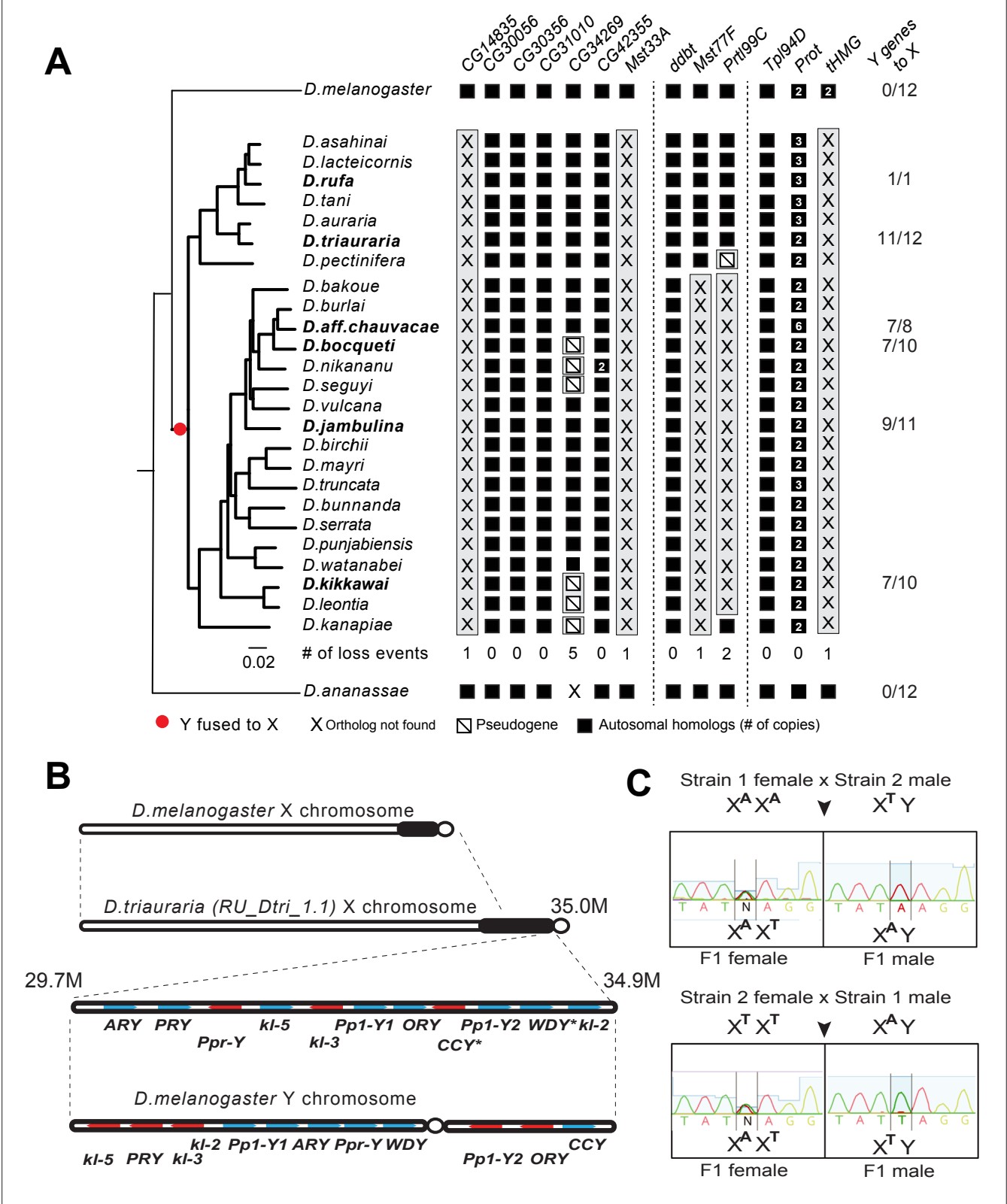

**Figure 6.** A dramatic loss of sperm nuclear basic protein (SNBP) genes coincided with a fusion of X and Y chromosomes in the *montium* group species. (**A**) Using a phylogeny of species from the *montium* group, we traced the retention or loss of SNBP genes that are otherwise primarily conserved across other *Drosophila* species. Genes retained in autosomal syntenic locations are indicated in black squares, whereas pseudogenes are indicated by an empty square with a diagonal line. We traced a total of 11 independent pseudogenization events. Three of these pseudogenization events occurred

*Figure 6 continued on next page*

*Figure 6 continued*

early such that all species from this group have lost *CG14835*, *Mst33A*, and *tHMG*. Three other SNBP genes were lost later (in some cases on multiple occasions) and are, therefore, missing only in a subset of species. For example, we infer that *CG34629* was lost on at least five independent occasions (and also in outgroup species *D. ananassae*). We correlated this dramatic loss of otherwise-conserved SNBP genes with the X-chromosome linkage of genes that are ancestrally Y-linked in other *Drosophila* species, shown on the right. For example, of 12 Y-chromosomal genes in most related species, including *D. melanogaster* and *D. ananassae*, most are now X-linked in *montium* group species (e.g., 11/12 in *D. triauraria*, 9/11 in *D. jambulina*, and 7/10 in *D. bocqueti* and *D. kikkawai*). We note these species still harbor a Y chromosome; however, this Y-chromosome lacks most ancestrally Y-linked genes. (**B**) We traced the chromosomal arrangement and linkage of ancestrally Y-linked genes in *D. triauraria* using new genome assembly (NCBI accession: GCA_014170315.2) and genetic crosses in (**C**). We were able to show that the *D. triauraria* X chromosome represents a fusion of the X chromosome (e.g., from *D. melanogaster*) and chromosomal segments containing 11 protein-coding genes that are typically found on the Y chromosome (e.g., from *D. melanogaster*). Genetic crosses confirmed the X-linkage of 9 of these previously Y-linked genes. The lack of allelic differences in *D. triauraria* prevented us from confirming this for the other two genes: *CCY* and *WDY*. (**C**) An example of the genetic cross used to verify X-linkage. Using genetic crosses between different *D. triauraria* strains with allelic variation in ancestral Y-linked genes, we evaluated whether male flies inherit these genes maternally, paternally, or from both parents. We observed only maternal inheritance, confirming the X-chromosomal linkage of these genes.

The online version of this article includes the following figure supplement(s) for figure 6:

**Figure supplement 1.** Phylogenetic analyses help distinguish between two models of relocation of ancestrally Y-linked genes.

genes, i.e., all except *JYalpha*, are located on the same region of the X chromosome (**Figure 6B**). The most parsimonious explanation for these findings is a single translocation fused most of the Y chromosome to the X chromosome in the ancestor of the *montium* group of species. Although *montium* group species still harbor a Y chromosome, their Y chromosome is missing most ancestrally Y-linked genes. After this ancient X-Y chromosomal fusion, some Y-linked genes, e.g., *PRY*, relocated back to the Y chromosome in some species (**Figure 6—figure supplement 1**).

We carried out genetic mapping studies to confirm our unexpected inference of a Y-to-X translocation in the *montium* group. For this, we performed genetic crosses between different *D. triauraria* strains harboring polymorphisms on ancestrally Y-linked genes (**Conner et al., 2021**). To unambiguously infer the chromosomal location of these genes, we focused on whether SNPs in these genes were paternally or maternally inherited. If the SNPs were Y-linked, we would expect them to be strictly paternally inherited, whereas we would expect both paternal and maternal inheritance in the case of autosomal linkage. In contrast to these expectations, all F1 males inherit alleles of ancestral Y-linked genes from their mother, unambiguously indicating their X-chromosomal linkage (**Figure 6C**). Our finding of an X-Y fusion via both computational and genetic linkage analyses is remarkable because X-Y fusion is an extremely rare evolutionary event that has only previously been documented in one vole species (**Couger et al., 2021**).

The translocation of a large segment of the Y-chromosome to the X-chromosome in the *montium* group would render any X-versus-Y meiotic drive encoded in this chromosomal region obsolete or costly. As a result, there would be active selection to jettison such meiotic drive systems on both the X and Y chromosomes. Indeed, no meiotic drive has been documented in the *montium* species (**Courret et al., 2019**), although this absence could simply reflect the inadequate number of studies for meiotic drive. Following an X-Y fusion, any autosomal SNBP genes required to suppress meiotic drive would become dispensable, leading to their degeneration. Thus, our hypothesis of genetic conflicts between sex chromosomes can explain the loss or degeneration of SNBP genes following alleviation of the sex-chromosomal conflict as well (**Figure 7**).

## Discussion

Our analyses of *Drosophila* SNBP genes reveal many similar patterns to rapid protamine evolution observed in mammals, including a pervasive trend of positive selection. However, there are also some dramatic differences, which may stem from distinct biological functions and selective pressures. One of the most dramatic differences is the evolutionary turnover of SNBP genes in *Drosophila* versus mammals. Most mammals share four SNBP genes: *TNP1*, *TNP2*, *PRM1*, and *PRM2*. In contrast, SNBP repertoires vary extensively between *Drosophila* species, with both dramatic gains and losses. We discovered several independent, species-specific amplifications of SNBP genes from 5 to >50 copies that preferentially occurred on sex chromosomes (**Figure 3** and **Table 2**). Conversely, we also found that several SNBP genes were lost in the *montium* group, coinciding with an X-Y chromosome fusion

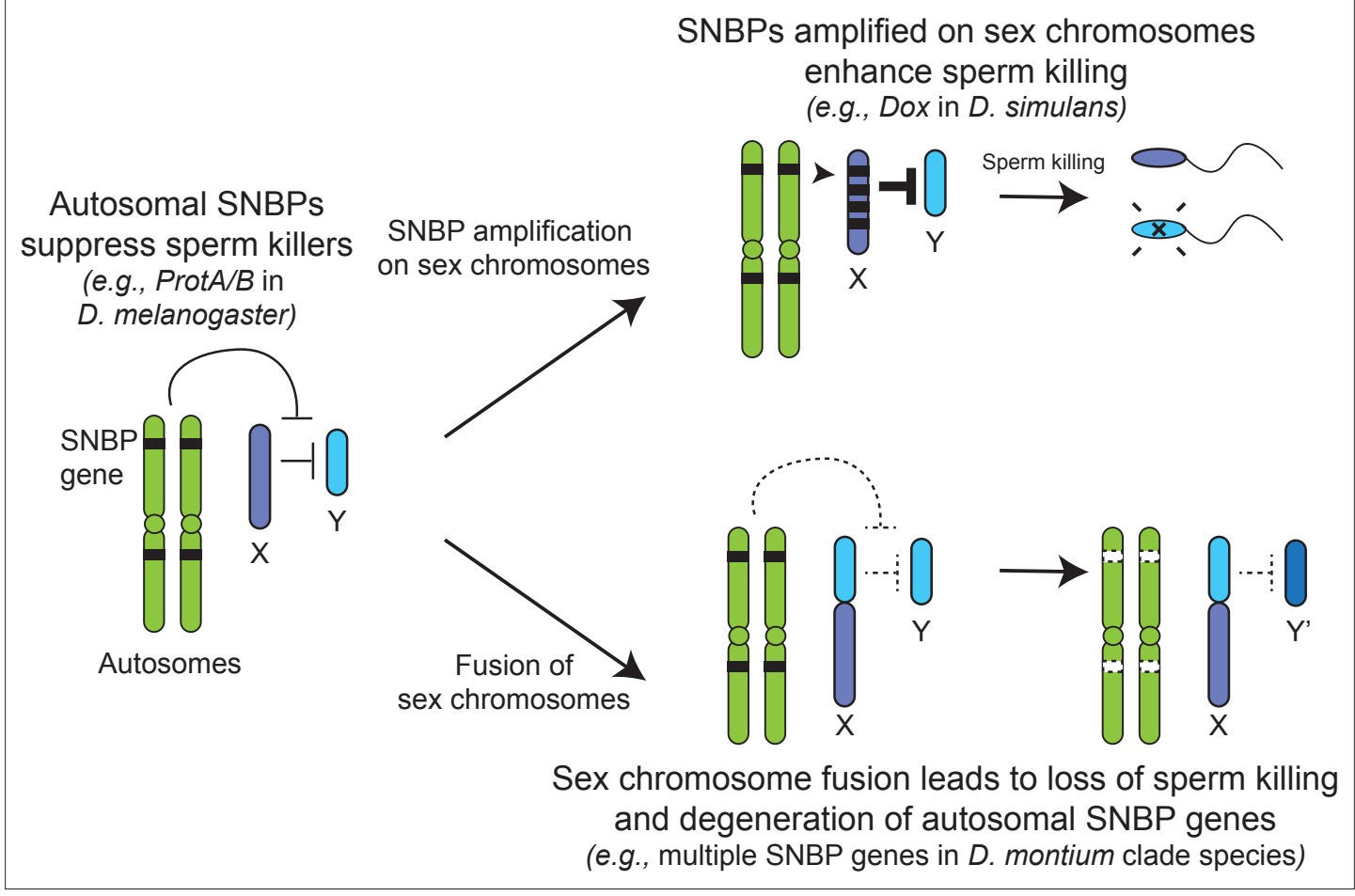

**Figure 7.** Genetic conflict between sex chromosomes may explain the rapid turnover of sperm nuclear basic protein (SNBP) genes in *Drosophila* species. SNBP genes are ancestrally encoded on autosomes where we hypothesize that some of them act to suppress meiotic drive between sex chromosomes (e.g., *ProtA/B*). However, in some cases, paralogs of these SNBP genes duplicate onto sex chromosomes where they undergo dramatic amplification. We propose that this amplification creates an opportunity for them to act as meiotic drive elements themselves (e.g., *Dox*), imbuing sex chromosomes that inherit them with transmission advantages. A fusion of the sex chromosomes (e.g., *D. montium* species group) leads to a loss of meiotic competition between sex chromosomes, which will subsequently lead to the loss or degeneration of the suppressing SNBP genes on autosomes since their drive suppression functions are rendered superfluous.

event (*Figure 6* and *Table 2*). In other lineages, three SNBP genes that became linked to sex chromosomes via chromosomal fusions either degenerated or translocated back to autosomes (*Figure 5*).

Based on all these observations, we hypothesize that many SNBP genes, such as the *ProtA/B* genes in *D. melanogaster*, arise and are retained on autosomes to act as suppressors of meiotic drive (*Gingell and McLean, 2020*). In contrast, sex chromosomal amplifications of SNBP genes, like the *Dox* amplification, act as meiotic drivers, potentially by disrupting the histone-to-protamine transition directly via their protein products or indirectly by affecting the expression or function of parental autosomal SNBP genes (*Vedanayagam et al., 2021*; *Muirhead and Presgraves, 2021*; *Tao et al., 2007a*). X-Y chromosome fusions eliminate the circumstances that facilitate meiotic drive and may lead to the degeneration of otherwise conserved SNBP genes, whose functions as drive suppressors are no longer required. Thus, unlike in mammals, sex chromosome-associated meiotic drive might be one of the primary causes of SNBP evolution in *Drosophila* species.

Why would meiotic drive only influence *Drosophila*, but not mammalian, SNBP evolution? One important distinction may arise from the timing of SNBP transcription. In *D. melanogaster*, most SNBP genes are transcribed before meiosis but translated after meiosis (*Jayaramaiah Raja and Renkawitz-Pohl, 2005*; *Rathke et al., 2007*; *Barckmann et al., 2013*). Thus, SNBP transcripts from a single allele, e.g., an X-linked allele, can be inherited and translated by all sperm, regardless of which chromosomes

they carry. Consequently, they can act as meiotic drivers by causing chromatin dysfunction in sperm without the allele, e.g., Y-bearing sperm. In contrast, mammalian SNBPs are only expressed post-meiosis (*Hecht et al., 1986*; *Peschon et al., 1987*) and, therefore, individual SNBP alleles can only affect the chromatin states of sperm in which they reside. Therefore, mammalian SNBP genes may be less likely to evolve as meiotic drivers.

Our findings do not rule out the possibility that forces other than meiotic drive might also be important for driving the rapid evolution and turnover of SNBP genes in *Drosophila* species. Indeed, we note that only some SNBP genes undergo both amplifications and losses across *Drosophila* species (*Figure 3* and *Table 2*). Other SNBP genes that do not experience such dramatic turnover still evolve rapidly (*Table 2*), including in *D. serrata*, a *montium* group species with a fused X-Y chromosome (*Table 1*). Signatures of positive selection that do not correspond to sex chromosomal meiotic drive could be instead explained by some form of haploid selection in spermatogenesis (*Raices et al., 2019*), such as sperm competition, and influence the evolution of sperm morphologies, as has been proposed in mammals.

Another possible selective pressure on SNBP genes may come from endosymbiotic *Wolbachia* bacteria. One means by which *Wolbachia* manipulate *Drosophila* hosts to ensure their preferential propagation is by mediating cytoplasmic incompatibility. Embryos produced between infected males and uninfected females fail to de-compact the paternal pronucleus and arrest in development (*Perreault, 1992*; *Loppin et al., 2000*). Recent studies have revealed that *Wolbachia* produces toxins that prevent the deposition of SNBP during spermatogenesis (*Kaur et al., 2022*) and delay SNBP removal in embryos post-fertilization to accomplish this cytoplasmic incompatibility (*Beckmann et al., 2019*; *Landmann et al., 2009*). Indeed, deletions of *ProtA/ProtB* exacerbate the intensity of cytoplasmic incompatibility imposed by *Wolbachia* in *D. melanogaster* (*Kaur et al., 2022*). Thus, selective pressure from *Wolbachia* toxins could also provide selective pressures for SNBP innovation.

Our analyses have focused on SNBP genes that can be readily identified because they possess HMG DNA-binding domains. However, this does not represent a comprehensive list of SNBP genes. Indeed, recent findings have shown that the *atlas* SNBP gene from *D. melanogaster* does not encode an HMG domain entirely and has arisen de novo (*Rivard et al., 2021*). Such genes might use an alternate means to bind DNA or may indirectly affect the localization of other SNBP proteins and therefore sperm chromatin. However, even focusing on the HMG-domain containing SNBP genes reveals an unexpected relationship between their essential function and evolution. SNBP genes encoding essential fertility functions arose recently and have been frequently lost in *Drosophila* species. In contrast, SNBP genes less essential for male fertility are more prone to evolve under positive selection. This suggests that SNBP genes with redundant roles in male fertility are more likely to acquire new function.

One clue for this unexpected lability emerges from discovering SNBP genes that are essential for *D. melanogaster* fertility but were lost in other species. For example, *ddbt* is a conserved SNBP gene with essential sperm telomere-capping function in *D. melanogaster*; loss of *ddbt* leads to defects in telomere capping and induction of telomeric fusions in embryos (*Yamaki et al., 2016*). We find that the two lineages that lost *ddbt*—all *D. willistoni* subgroup species, and *D. albomicans* in the *D. immigrans* species group—have a pair of neo-sex chromosomes due to the fusion of Muller element D and sex chromosomes (*Sturtevant and Novitski, 1941*; *Chang et al., 2010*; *Zhou et al., 2012*; *Ellison and Bachtrog, 2019*). We hypothesize that the loss of *ddbt* might have led to these chromosome fusions that cause the independently evolved neo-sex chromosomes. Alternatively, since telomeric sequences rapidly evolve in an 'arms-race'-like dynamic with telomere-binding proteins across *Drosophila* species (*Saint-Leandre and Levine, 2020*), particular chromatin rearrangements may have obviated the essential function of *ddbt* in some species, as has been hypothesized for other paternal-effect lethal chromatin genes (*Levine et al., 2015*). The binding specificity and function of *ddbt* also hint that other SNBPs might also bind to specific genomic regions and have other important biological roles, e.g., gene regulation and protecting against DNA damage (*Rathke et al., 2010*). This suggests that the nature of essential SNBP fertility functions can itself be idiosyncratic and species-specific, arising from the underlying rapid evolution of chromatin, as has been hypothesized for evolutionarily young but essential genes involved in centromere and heterochromatin function (*Kasinathan et al., 2020*; *Ross et al., 2013*).

## Materials and methods

### Molecular evolutionary analyses of SNBP genes

We use PAML 4.9 (*Yang, 2007*; RRID:SCR_014932) to calculate protein evolution rates (dN/dS) of SNBP genes and conduct site-model, branch-model, and branch-site model analyses. We compare the protein evolution rates of SNBP genes to the genome-wide rates (8521 genes) from the 12 *Drosophila* genomes project (*Clark et al., 2007*). Many SNBP genes analyzed in our study were missed in previous analyses because they are rapidly evolving and hard to align. Therefore, we aligned the coding sequences of orthologous SNBP genes from the same six *Drosophila* species used to calculate dN/dS previously (*Clark et al., 2007*), taking care to only use orthologous sequences. For all phylogenic analyses, we first constructed maximum-likelihood trees using iqtree 2.1.3 (*Minh et al., 2020*) using parameters '-m MFP -nt AUTO -alrt 1000 -bb 1000 -bnni'. We then calculated protein evolution rates of SNBP genes using the same parameters with the generated gene trees (model = 0 and CodonFreq = 2).

For the site-model codeml analyses, we analyzed 9–17 unambiguous orthologs from species in the *melanogaster* group to increase the power and accuracy. We compared NS sites models M1a to M2a, and M7 or M8a, to M8 using likelihood ratio tests to ask whether models allowing a class of sites where dN/dS exceeds 1 (e.g., M8) provide a better fit to the data than models that disallow positive selection (e.g., M7, M8a). We also used different codon parameter values (CodonFreq = 0, 2, 3) in our analyses to check whether our results were robust.

For the branch-model codeml analyses, we first simplified the tree by reconstructing the ancestral sequences of ampliconic genes in each species using MEGAX (10.1.8) (*Stecher et al., 2020*). We then compared models by assigning different protein evolution rates to different branches using PAML (CodonFreq = 2) and likelihood ratio tests. We tested several models, starting with the null model, which models the same protein evolutionary rate across branches (model = 0). We compared this null model to alternative models in which all X-linked branches share a common protein evolutionary rate that is different from other branches (model = 2). Finally, we compared a model in which the early duplication branches on both the X-linked copies and the parental copy share different protein evolutionary rates than in other branches (model = 2). We found that this latter model, with a higher protein evolution rate at early duplication branches, fits best across all models, so we applied this setting to conduct the branch-site test. We compared two models with all sites sharing the same protein evolution rate (fix_omega = 1) and various evolution rates (fix_omega = 0).

To look for positive selection in two individual lineages, *D. melanogaster* and *D. serrata*, we applied McDonald–Kreitman tests to compare within-species polymorphism and between-species divergence (*McDonald and Kreitman, 1991*). We used *D. simulans* as the closely related outgroup species for the *D. melanogaster* analysis, and *D. bunnanda* for the *D. serrata* analysis. To polarize the changes in each lineage, we inferred ancestral sequences of *D. melanogaster* and *D. simulans* using seven species in the *D. melanogaster* group (*D. melanogaster*, *D. simulans*, *D. mauritiana*, *D. sechellia*, *D. yakuba*, *D. erecta*, and other well-aligned outgroup species: *D. biarmipes*, *D. elegans*, *D. eugracilis*, *D. ficusphila*, or *D. rhopaloa*) using MEGAX (10.1.8; RRID:SCR_000667; *Stecher et al., 2020*). Similarly, we used five species in the *montium* group (*D. serrata*, *D. bunnanda*, *D. birchii*, *D. truncata*, and *D. mayri*) to polarize the *D. serrata*-specific changes. We extracted population data from public datasets of >1000 *D. melanogaster* strains (*Hervas et al., 2017*; *Lack et al., 2016*) and 111 *D. serrata* strains (*Reddiex et al., 2018*). We conducted both polarized and unpolarized McDonald–Kreitman tests using R scripts (https://github.com/jayoung/MKtests_JY) and confirmed our findings using an online server (*Egea et al., 2008*; http://mkt.uab.es/mkt/MKT.asp).

### Searching for homologs of SNBP genes in *Drosophila* and outgroup species

We used tblastn and reciprocal blastx to search homologs of all SNBP genes across all genome assemblies (RRID:SCR_004870; *Altschul et al., 1990*) using *D. melanogaster* protein sequences as queries. Since SNBP genes are rapidly evolving, we used the following parameters: e-value < 1e-2, amino acid identity > 20%, and blast score > 10. We further required that the best reciprocal blastx hit when searching *D. melanogaster* genes was the original query gene to ensure that we were recovering true orthologs. To further confirm questionable orthologs, e.g., only one species with the homolog in the lineage, we examined its synteny, anticipating that orthologs should also have shared syntenic

contexts. We also examined the syntenic regions of SNBP genes and conducted blastx using a lower threshold (e-value < 1) to confirm the loss of SNBP genes in some lineages, especially in the *montium* group species.

We used the abSENSE package (RRID:SCR_023223; *Weisman et al., 2020*) to calculate the probability of not detecting homologs in more diverged species (using E-value = 1), enabling us to distinguish whether our inability to detect homologs was due to rapid divergence or true absence.

## Transcriptomic analyses of SNBP genes

We combined the public gene annotations with our own annotated SNBP gene annotations and mapped publicly available transcriptome datasets (*Supplementary file 13*) to the genome assemblies using HiSAT2 (v2.2.1 with parameters –exon and –ss to specify the exon positions and splice sites; RRID:SCR_015530; *Kim et al., 2019*). We then estimated the expression levels using the gene annotations as input for Stringtie (v2.1.4 with parameters -dta -G to specify annotation files; RRID:SCR_016323; *Kovaka et al., 2019*). For single-cell transcriptomic data, we downloaded and used the scripts and data from (*Witt et al., 2021*). We modified the published scripts to extract and plot the expression level of SNBP genes using the NormalizeData function of Seurat (RRID:SCR_007322; *Hao et al., 2021*).

## Assigning sex-linkage of contigs in genome assemblies

We mapped Nanopore and Illumina reads from male samples (sometimes from mixed-sex or female samples) to genome assemblies using minimap2 (RRID:SCR_018550; *Li, 2021*) and bwa-mem (RRID:SCR_022192; *Vasimuddin et al., 2019*) using the default parameters. We calculated coverage of each site using samtools depth and estimated the median coverage of each 10 kb window and each contig. We then examined the genome-wide distribution of 10 kb window coverage using the density function in R. We called two peaks of coverage using turnpoints function in R. As we expected, the prominent peak with higher coverage mostly represents autosomal regions, whereas the lower coverage peak mostly represents sex-linked regions. We used the average of coverage from the two peaks as the threshold to assign autosomal and sex-chromosomal contigs.

We also confirmed our assignments using the sequence-homology method. We identified orthologs of 3285 highly conserved genes in all species using BUSCO v5.0.0 with default setting and diptera_odb10 database (RRID:SCR_015008; *Manni et al., 2021*). Since the X chromosome is conserved across *Drosophila* species, we identified orthologs of X-linked genes to assign X-linked contigs. Lastly, for species with Illumina data from both males and females, we distinguished X from Y chromosome-linked contigs using a previously described method (*Chang and Larracuente, 2019*). We could not reliably distinguish their sex chromosomes from autosomes in *Lordiphosa* species (probably due to high heterozygosity or lower assembly quality) and therefore could not confidently assign the chromosomal location of their SNBP amplifications.

We used genetic mapping to examine the X-linkage of ancestral Y-linked genes in *D. triauraria*. We first identified polymorphic SNPs in *D. triauraria* SNBP genes using publicly available Illumina data (*Conner et al., 2021*) and called SNPs (bcftools call -m -Oz; RRID:SCR_005227) in each strain (*Li et al., 2009*; *Li, 2011*). We took advantage of the fact that F1 males will only inherit maternal X alleles. Four *D. triauraria* strains (14028-0691.01 [National *Drosophila* Species Stock Center], KMJ1 [Ehime-Fly: E-15304], OKNG12-6 [Ehime-Fly: E-15309] and YKS-MTK [Ehime-Fly: E-15303]) were kindly provided by Dr. Brandon Cooper. We crossed two strains with different alleles of ancestrally Y-linked genes (14028-0691.01 with KMJ1 and OKNG12-6 with YKS-MTK) and genotyped these genes in F1 males using PCR (*Supplementary file 14*) and Sanger sequencing to examine whether their allele-specific inheritance patterns were more consistent with Y-linkage (paternal inheritance), X-linkage (maternal inheritance), or autosomal linkage (both paternal and maternal inheritance).

## CRISPR/Cas9 knockout and fertility assays

To generate the *CG30056* knockout strain, we first cloned two guide RNAs, targeting either the 5' or 3' end of *CG30056*, into pCFD4-U6:1_U6:3tandemgRNAs (RRID:Addgene_49411) using NEBuilder HiFi DNA Assembly Master Mix (NEB catalog E2621). For the repairing construct, we used independent PCR reactions to amplify 3xP3 DsRed, the backbone of pDsRed-attP (RRID:Addgene_51019), and ~1 kb homologous sequences of upstream and downstream of *CG30056* using

PCR independently. These four fragments were annealed using NEBuilder HiFi DNA Assembly Master Mix. We then used the Q5 Site-Directed Mutagenesis Kit (NEB catalog E0554S) to mutate the gRNA target PAM sites on the repairing construct so that they would not also be targeted by the guide RNAs we used. All primers are listed in *Supplementary file 14*. These two constructs were injected into y[1] M{GFP[E.3xP3]=vas-Cas9.RFP-}ZH-2A w[1118] (RRID:BDSC_55821) embryos by BestGene. The resulting transgenic flies were backcrossed to *yw*, selected using the DsRed marker, and balanced by *CyO*. The transgenic flies were further confirmed by PCR using independent primer sets (*Figure 3A* and *Supplementary file 14*).

For comparisons, we also tested male fertility effects of two SNBP genes previously shown to be essential (*Mst77F*) or not (*Tpl94D*) for male fertility. We obtained *Mst77F* (Δ1) knockout flies (kindly provided by Dr. Benjamin Loppin *Kimura and Loppin, 2016*) and two fly strains carrying large deletions spanning *Mst77F* (RRID:BDSC_24956 and RRID:BDSC_27369). For *Tpl94D*, we used a mutant from the *Drosophila* Gene Disruption Project (RRID:BDSC_26333; *Bellen et al., 2004*) and a fly carrying a large deletion spanning this gene (BDSC_7672; *Parks et al., 2004*).

We obtained males for fertility assays by crossing female virgin transgenic flies with other transgenic flies or flies carrying a large deletion (RRID:BDSC_30585). To measure their fertility, each resulting 2–5-day-old F1 male was crossed to two 2–5-day-old virgin females from the wildtype Oregon R *D. melanogaster* strain (kindly provided by Dr. Courtney Schroeder; RRID:BDSC_5 and *Wolbachia* cured) at room temperature. We transferred the mating pairs to new vials every three days and counted all resulting offspring from the first 12 days.

To detect subtle fertility defects in *CG30056* knockout flies, we replaced the balancer chromosome with the wildtype chromosome from the strain used for the CRISPR experiment (RRID:BDSC_55821). We also performed a sperm exhaustion assay in which we crossed each 2–5-day-old F1 male to ten 2–5-day-old virgin females from the wildtype Oregon R *D. melanogaster* strain at 25°C. We transferred the mating pairs to new vials every 3–4 days and counted all resulting offspring from the first 14 days. We excluded crosses that produced less than 20 offspring at the end, except those from *Mst77F* mutants. Most of these crosses did not have progeny because they did not mate well. Since *Mst77F* mutants are sterile (*Kimura and Loppin, 2016*), we cannot exclude such data.

## Acknowledgements

We thank Tobias Warnecke and two anonymous reviewers for their comments and suggestions to improve the manuscript. We are grateful to Cécile Courret, Janet Young, Phoebe Hsieh, Pravrutha Raman, and Rupinder Kaur for their comments on the manuscript. Finally, we thank Barbara Wakimoto, Amanda Larracuente, John Sproul, and members of the Malik, Ahmad, and Henikoff labs for fruitful discussions. We also thank Benjamin Loppin, Brandon Cooper and the Bloomington Drosophila Stock Center (supported by NIH P40OD018537) for the *Drosophila* strains used, and acknowledge Flybase (supported by NHGRI Award U41HG000739) for helping build bioinformatic tools across various *Drosophila* species' genomes. We are supported by a Damon-Runyon Cancer Research Foundation postdoctoral fellowship DRG 2438-21 (to C-HC) and National Institutes of Health grant R01-GM74108 (to HSM). The funders played no role in the study design, data collection and interpretation, or the decision to publish this study. HSM is an Investigator of the Howard Hughes Medical Institute. This article is subject to HHMI's Open Access to Publications policy. HHMI lab heads have previously granted a nonexclusive CC BY 4.0 license to the public and a sublicensable license to HHMI in their research articles. Pursuant to those licenses, the author-accepted manuscript of this article can be made freely available under a CC BY 4.0 license immediately upon publication.

## Additional information

### Funding

| Funder | Grant reference number | Author |
|---|---|---|
| Damon Runyon Cancer Research Foundation | DRG 2438-21 | Ching-Ho Chang |
| National Institute of General Medical Sciences | R01-GM74108 | Harmit S Malik |
| Howard Hughes Medical Institute | | Harmit S Malik |

The funders had no role in study design, data collection and interpretation, or the decision to submit the work for publication.

### Author contributions

Ching-Ho Chang, Conceptualization, Data curation, Formal analysis, Funding acquisition, Validation, Investigation, Visualization, Writing – original draft, Project administration, Writing – review and editing; Isabel Mejia Natividad, Investigation, Writing – review and editing; Harmit S Malik, Conceptualization, Supervision, Funding acquisition, Visualization, Writing – original draft, Project administration, Writing – review and editing

### Author ORCIDs

Ching-Ho Chang http://orcid.org/0000-0001-9361-1190
Harmit S Malik http://orcid.org/0000-0001-6005-0016

### Decision letter and Author response

Decision letter https://doi.org/10.7554/eLife.85249.sa1
Author response https://doi.org/10.7554/eLife.85249.sa2

---

## Additional files

### Supplementary files

• Supplementary file 1. Probability that rapid evolution has obscured homolog detection of young SNBP genes in *Drosophila* species. We conducted abSENSE analyses (*Weisman et al., 2020*) using the blast scores in related species with detected orthologs and inferred likely Blast scores of the orthologs in more related species given the divergence of species. Then we estimated the probability of failing to detect a homolog (if it were present) in species of various divergence levels (using E-value = 1).

• Supplementary file 2. Expression levels of SNBP genes in *Drosophila* and *Scaptodrosophila* species. We estimated the expression levels (using TPM) of SNBP genes using publicly available transcriptome datasets of different tissues (*Supplementary file 12*). The data is also illustrated in *Figure 1—figure supplement 3*.

• Supplementary file 3. Sequence information of SNBPs *Drosophila* species. We collected the sequences of SNBPs and their homologs from the NCBI database. We calculated the isoelectric point and length of each protein using Geneious 2022.1.1 (https://www.geneious.com).

• Supplementary file 4. Evolutionary rates of SNBP genes in *D. melanogaster* subgroup species. We used PAML to estimate evolutionary rates of SNBP genes using the same parameters and the same six *Drosophila* species used in the 12 *Drosophila* genomes project (*Clark et al., 2007*). For comparison, we used the evolutionary rates of other genes from the 12 *Drosophila* genomes project (*Clark et al., 2007*).

• Supplementary file 5. McDonald–Kreitman test results for SNBP genes in *D. melanogaster* and *D. serrata*. We looked for positive selection in two lineages, *D. melanogaster* and *D. serrata*, using McDonald–Kreitman tests to compare within-species polymorphism to between-species divergence (*McDonald and Kreitman, 1991*). We used *D. simulans* as the closely related species for the *D. melanogaster* analysis and *D. bunnanda* for the *D. serrata* analysis.

• Supplementary file 6. No evidence for positive selection on SNBP genes using the site model in

PAML in *D. melanogaster* subgroup species. We aligned 9–17 unambiguous orthologs from species in the *D. melanogaster* group to test whether a subset of sites evolves under positive selection. We compared NSsites models M1a to M2a, and M7 or M8a to M8 using likelihood ratio tests. We ran each model using several codon parameter choices (CodonFreq = 0, 2, 3) to check whether the results were robust. For example, *CG30056* shows a signal of difference selection strength across sites using CodonFreq = 2 (p=0.0003), but not CodonFreq = 0 or 3 (p=1 and 0.16, respectively).

• Supplementary file 7. Low frequency of inactivating polymorphisms in SNBP genes from *D. melanogaster* populations. We extracted population data using an available dataset of >1000 *D. melanogaster* strains (*Hervas et al., 2017*; *Lack et al., 2016*) and long-read assemblies, and documented inactivating mutations in SNBP genes. We found that loss-of-function variants of SNBP genes segregate at very low frequencies (<1%) among *D. melanogaster* strains. The only exceptions are *CG14835* (1.5% frequency of frameshift mutation in worldwide populations), *tHMG1* (5.4% frequency of deletion based on long-read assemblies), and *ddbt* (1.2% frequency of loss of start codon in non-African populations). However, the *ddbt* mutation is likely to be benign owing to an alternate start codon just a few codons downstream of the canonical start site. In contrast, variants that are not likely to impair function (small in-frame indels) can segregate at higher frequency, e.g., a 15 bp insertion variant of *tHMG2* is present at 70% frequency in worldwide *D. melanogaster* populations. This suggests nearly strict retention of all SNBP genes, whether they were shown to be essential for male fertility in laboratory experiments or not, in all sequenced strains of *D. melanogaster*.

• Supplementary file 8. The fertility assays of SNBP knockout and mutated flies. We performed fertility assays comparing *CG30056* homozygous knockout flies with heterozygous controls. We also assayed fertility of knockout strains for the fertility-essential *Mst77F* gene, and the fertility-nonessential *Tpl94D* gene, together. We also documented the sex-ratios of the resulting progeny in *Figure 2*. Consistent with previous findings, we found that *Mst77F* knockout males are essentially sterile and *Tpl94D* knockout males were indistinguishable from their heterozygous controls. We found either no or weak evidence of fertility impairments in three different crosses with homozygous *CG30056* knockout males. We observe no significant evidence of sex-ratio distortion that would suggest an X-versus-Y meiotic drive in progeny resulting from either *CG30056*, *Mst77F,* or *Tpl94D* knockout males.

• Supplementary file 9. Chromosomal assignments for each contig containing SNBP genes. We assigned the location of SNBP-containing contigs using synteny (Muller elements) and coverage analysis. We used BUSCO genes on these contigs to assign their most likely location on Muller elements. We also mapped available male Nanopore or Illumina reads to the assemblies and estimated coverage on the contigs compared to autosomal contigs. If the normalized read coverage is significantly less than 1, we assign the contigs to either X or Y chromosome.

• Supplementary file 10. The copy number and chromosome location of SNBP homologs using BLAST in each species. We summarized the data from *Supplementary file 8* and also manually curated data from some amplified SNBP genes using extra assemblies or Illumina reads (shown in red). To determine the chromosomal location of some amplified SNBP genes, we mapped male and female Illumina reads from different resources to the assemblies of 10 species (*Supplementary file 13*). This allowed us to assign contigs to the X or Y chromosome unambiguously. For *D. melanogaster* and *D. simulans*, we used assemblies with better contiguities (GCA_000778455.1 [*Krsticevic et al., 2015*] and GCA_004382185.1 [*Chakraborty et al., 2021*]).

• Supplementary file 11. PAML analyses reveal different selection forces in *tHMG* duplicates of *D. simulans* clade species. We analyzed *tHMG* copies from *D. simulans* clade species to infer their selective pressures (*Figure 4*). We compared branches with different protein evolution rates using likelihood ratio tests (CodonFreq = 2). Our models include the null model (same protein evolution rate across branches), a model where all X-chromosome branches share a rate that is different from the rate on all other branches ('all X'), and a model where the early duplication branches on both the X-linked copies and the parental copy share a rate that is different from all other branches ('Duplication'). We compared two models with all sites that share the same protein evolution rate (fix_omega = 1) and various evolution rates (fix_omega = 0), and did not find evidence of positive selection. The duplication model fits best across all models, so we also used this model to conduct a branch-site test. No evidence for positive selection was using the branch-site test.

• Supplementary file 12. Location and degeneration of SNBP genes in species with neo-sex chromosomes. We report the chromosomal locations of each SNBP gene in species with neo-sex chromosomes illustrated in *Figure 5*.

• Supplementary file 13. Sequence data resources and information used in this study.

- Supplementary file 14. Primer sequences used in this study.
- MDAR checklist

## Data availability

All data generated or analyzed during this study are included in the manuscript and supporting file; Source Data files have been provided for all Figures. No sequence data have been generated for this manuscript.

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
