## [Editor Report]

Chang et al. used a previously published set of highly contiguous genomes to infer the drivers of the evolution of sperm nuclear basic proteins and find several instances of gene duplication mainly occurring in sex chromosomes. Moreover, they provide a genetic characterization of one such protein (CG30056). The paper was initially reviewed by experts in the field through Review Commons. The three reviewers were enthusiastic about the potential of the paper and made a set of suggestions to make the paper stronger. The authors incorporated the suggestions when relevant, added a new (and relevant) experiment, edited the manuscript as requested, and clarified some instances that could be further developed.

---

## [Decision Letter]

[Editors' note: this paper was reviewed by Review Commons.]

---

## [Author Response]

Overall comments

We are pleased by the reviewers’ comments and appreciate their suggestions for improvements. In addition to correcting small typos throughout the manuscript, the major changes we did in response to their comments are as follows:

Changed the title of our paper to reflect the strong evolutionary correlation more accurately between sex chromosomal meiotic drive and gains/losses of SNBP genes in *Drosophila.*New experiments to test the role of the well-conserved, universally retained SNBP, *CG30056,* in male fertility in *D. melanogaster*. Although reviewers had suggested we could eliminate this section, we felt that this would add a lot of weight to the unexpectedly inverse relationship between age/retention and fertility functions of SNBP genes. Thus, over the past few months, we have carried out new experiments with increased sample sizes, better controls, and sperm exhaustion. These new results strengthen our earlier analyses.Better clarification of the X-Y chromosome fusion, which is a new observation, in the *montium* group via careful rewriting as partly suggested by Reviewer #2.Highlighting that the genetic conflicts hypothesis does not rule out a role for sperm competition or other conflicts in shaping SNBP evolution in a revised Discussion.

All changes in response to the reviewer’s comments have been detailed in our point-by-point response (below). You will see that we have addressed almost all the suggestions made, including with new experiments. The only reviewer suggestions (all optional from Reviewer 3), which we did not directly address in our revision are:

Branch specific protamine evolution analyses for sex chromosome amplified SNBP genes: given the state of SNBP gene annotation and the difficulties of assembling these genes in large tandem arrays, this will require considerable work and is beyond the scope of the paper.Covariation between SNBP evolution and sperm morphology: We cannot perform these experiments as there is a paucity of sperm morphology data currently. Obtaining this data reliably is a significant undertaking.Are SNBP genes more prone to be lost than average in the *montium* group: We have not comprehensively examined all loss events in the *montium* group or any other *Drosophila* group. This is also a non-trivial analysis, albeit it would be very interesting. However, we believe the more relevant comparison is whether these lost SNBP genes are more likely to be retained in non-*montium* species, which they are, as we now highlight.

We hope you will favorably judge our good faith efforts to address all other reviewers’ comments, and their laudatory comments during the previous round of reviews.

Reviewer #1 (Evidence, reproducibility and clarity (Required)):Chang and Malik present a comprehensive evolutionary analysis of sperm nuclear basic proteins (SNBPs) in *Drosophila*. In addition, they provide a preliminary functional characterization of one such protein (CG30056) and describe a newly discovered X-Y chromosomal fusion in the *Drosophila* montium species group. All of these findings are interesting and important, but the headline from this study is the well-supported possibility that SNBPs, or at least a large fraction of them, function in suppressing X vs. Y chromosome meiotic drive. While this hypothesis is challenging to test experimentally, the authors provide strong correlational evidence that SNBPs are associated with drive by documenting these proteins' rapid evolution. This rapid evolution takes the form of sequence changes (as predicted by coevolution between drivers and suppressors of drive), gene amplification in cases when SNBPs move to sex chromosomes (consistent with the SNBP becoming a potential agent of drive for its new "home chromosome"), and gene loss in species with X-Y chromosome fusions (in which drive is not predicted to occur).Overall, this is an excellent, comprehensive study. The phylogenetic and genomic analyses are first- rate (and one of the first to make use of the new 101 *Drosophila* genomes); the logic is very well explained; conclusions are supported by multiple lines of evidence; the writing and figures are clear and accessible; and, the findings are fascinating. It's a good sign that it is easy to imagine several experiments one could do to follow up on this study, but I do not feel any are required in revision, as the manuscript is comprehensive as is. Thus, I have just a few minor points the authors may wish to consider in making revisions and a few suggestions for clarity/typos.

We thank the reviewer for their positive comments on our work.

1) I would be interested in whether the authors think that all SNBPs in a given *Drosophila* species function(ed) in meiotic drive, or whether some fraction may play other roles, such as sexual selection or chromatin compaction, which have been the traditional hypotheses for SNBP function. Relatedly, given the high turnover of SNBPs the authors observe and the fact that some melanogaster-essential SNBPs are younger genes, would they like to comment on whether the subsets of SNBPs involved in drive/suppression vs. chromatin packaging/sperm traits/Wolbachia defense are likely to differ from across fly species?

The reviewer raises an excellent point. In our revised discussion, we now speculate that different SNBPs might have distinct functions. For example, the same subset of SNBPs is subject to gene amplification and loss whereas other SNBPs are subject to less turnover. Moreover, even this stable set of SNBPs evolves rapidly, including in the *montium* group of species that have undergone dramatic SNBP loss. As the reviewer suggests, sperm competition or pressures from Wolbachia toxins might be is a driving force for sperm evolution. We discuss these possibilities and conclude in our discussion: “Our findings do not rule out the possibility that forces other than meiotic drive are also important for driving the rapid evolution and turnover of SNBP genes in *Drosophila* species.”

2) What do the authors make of the lower isoelectric points for a few of the SNBPs (e.g., CG31010 with π = 4.77 in Table 1)? These proteins have identifiable HMG box domains, so is the π driven lower by other parts of the protein sequence?

We thank the reviewer for raising this point. We found that the π of HMG domains can range from 6 to 12. Thus, the π is driven by both HMG domains and other parts of proteins. We now include the π of the whole SNBP protein and the HMG domain alone in Table 1. We do not have enough biochemical information to speculate on how these differences could alter SNBP function.

3) For readers less familiar with the field, it may help to spell out (e.g., on p. 6) why the authors consider ProtA/B to be important for fertility. Some of the previous papers on these genes describe them as dispensable – though the present authors are correct that these previous studies do detect fertility defects of various magnitudes under some conditions.

We agree with the reviewer. Previous studies are in disagreement about the importance of *ProtA/ProtB* for male fertility- while no significant effects were seen under standard fertility assays, sperm exhaustion conditions (mating with excess females) did reveal fertility effects. We have now added these references and discussed *ProtA/ProtB* more fully in our revision.

4) On p. 9, paragraph 2, the data showing that "six different SNBP genes underwent 11 independent degeneration events in the montium group" are shown in Figure 6A, not 5A.

Thank you. This has been fixed in our revision.

5) The summary Table 2 is useful, but I wonder whether including relative levels of expression and dN/dS in addition to ordinal rankings might help clarify. For instance, if there were a drop off in mean expression level between the 5th and 6th most highly expressed SNBP, this wouldn't be evident from the table.

We agree with this suggestion and have now added this information.

6) In Figure 3, I like the use of the clean CG31010 figure in panel A to illustrate the circular representations. In addition, though, it might be useful to show Prot's graph at this same, larger size, since it's the most complicated and will likely be most closely examined.

We agree with this suggestion and have now amended this figure in line with the reviewer’s suggestion.

7) In Figure 4, the end of the legend says that the species tree is shown "on the right," but it's on the left in the figure.

Thank you. This has been fixed in our revision.

Cross-Consultation CommentsI agree with both Reviewers 2 and 3 that the title could be changed to be a bit more tentative. I'd had this thought as well.

We agree with this suggestion. We have now amended this title to “Expansion and loss of sperm nuclear basic protein genes in *Drosophila* correspond with genetic conflicts between sex chromosomes.”

I agree with Reviewer 2 that the fertility assay could be conducted with a larger sample size and a better control in order to be better compared with how the authors described other published fertility phenotypes for SNBPs. For the control, crossing the deletion line to y w (or w1118) and using the resulting heterozygotes (KO/+) would be better than using the mutation over the balancer chromosome (KO/CyO).

We agree with both suggestions. We now compare fertility between KO/KO and KO/+ males in sperm exhaustion assays. Our more stringent fertility assays find no evidence of *CG30056* role in male fertility, strengthening our previous findings. We have now added the motivation for the new assays and the new results to our Revision.

I agree with Reviewer 3's third bullet point about spending a bit more time on the different possible roles that SNBPs could play in spermatids. (This is a more eloquent version of my review point #1.)

We have now expanded our discussion of other possibilities in our revision.

I agree in principle with Reviewer 3's first bullet point about examining whether SNBP evolution correlates with changes in sperm morphology, but this feels like it could be a whole, fascinating study on its own, while this manuscript is already packed with data. I'd welcome the authors' thoughts about this in discussion, but wouldn't personally require a formal analysis of this to be added prior to publication.

We also agree that this would be an interesting test. However, we are not able to do the test due to the scarcity of sperm phenotype data in *Drosophila*. We also think that our original version unintentionally downplayed this possibility. Our revised discussion makes clear that the rapid evolution of some *Drosophila* SNBP genes may be driven by sperm competition, just as in mammals, and influence the evolution of sperm morphologies.

Reviewer #1 (Significance (Required)):This study describes an important conceptual advancing in our understanding of the evolution and potential functions of sperm nuclear basic proteins (SNBPs) in *Drosophila*, which stands in interesting contrast to the functional roles of equivalent proteins in primates. It should be of broad interest to biologists studying spermatogenesis, meiotic drive, and genome evolution, both in and out of *Drosophila*.

We thank the reviewer for their positive appraisal.

To contextualize the work, paternal DNA is typically compacted during spermatogenesis. This process involves the replacement of histones with other small, positively charged proteins in a sequential order, ending with protamines that bind DNA in mature sperm. In *Drosophila*, work over the last two decades (largely from the labs of R. Renkawitz-Pohl, B. Loppin and B. Wakimoto) has identified more than a dozen sperm nuclear basic proteins that localize to condensing/condensed spermatid nuclei.Two interesting observations have been that many of these proteins are dispensable for male fertility, and the proteins vary in their degree of evolutionary conservation. Recent work from Eric Lai's lab (J Vedanayagam et al. 2021, Nat Ecol Evol) showed that in D. simulans and sister species, at least one of these SNBP genes (Prot) underwent gene amplification and now acts in those species as a meiotic driver. This finding suggested the hypothesis, tested thoroughly in the present study, that the rapidly evolving SNBP gene family could be involved in causing or suppressing meiotic drive. Consistent with this idea, the authors here find that SNBP genes expand in copy number more frequently when they move from autosomes to sex chromosomes (consistent with the idea that they may cause or contribute to drive), and that otherwise well-conserved SNBP genes are lost in a group of species in which sex chromosome meiotic drive is not expected to occur. These findings are based on a thorough and well conducted phylogenomic and molecular evolutionary analysis of SNBPs across dozens of *Drosophila* species. Overall, this work generates exciting new hypotheses about the function of SNBPs and should be widely read both within and outside of the field.

We are grateful for the reviewer’s accurate summary of our work and its significance. We share the reviewer’s excitement and expect that more studies will explore the new function of SNBPs in multiple taxa soon.

Keyword describing my field of expertise: *Drosophila*, molecular evolution, reproduction, genetics, genome evolution.Reviewer #2 (Evidence, reproducibility and clarity (Required)):The paper describes interesting patterns on the evolution of *Drosophila* SNBP genes, and proposes a very interesting explanation, namely, that meiotic drive is the main evolutionary force behind these patterns. Some of these observations have recently been made by other authors in a single case (the Dox genes in D. simulans), but not in the scale and breadth of the present manuscript. The manuscript combines an extensive investigation of available genomes with expert analysis, and new experimental data. In particular, the finding that the ancestral Y became incorporated into de X in montium species is very exciting, and may provide a smoking gun for the explanation proposed by the authors. Overall, I think it is a very good paper. I do have several criticisms and suggestions that may help to improve it.

We are grateful for the positive comments of the reviewer and for their constructive criticism and suggestions, which we have incorporated into our revision.

The paper has a speculative side that it almost unavoidable given its novelty and breadth. I do not see this as a problem per se, but I think the uncertain/unsupported/problematic points should be more openly presented to the readers. The main cases I noted are:1) The title of the manuscript states that "Genetic conflicts between sex chromosomes drive expansion and loss of sperm nuclear basic protein genes in *Drosophila*", but the evidence is somewhat circumstantial, and the patterns may be explained also by other known phenomena (e.g., demasculinization of the sex chromosomes; below). I think the tone of the end of the Introduction reflects more faithfully the strength of the evidence ("Thus, we conclude that rapid diversification of SNBP genes might be largely driven by genetic conflicts between sex chromosomes in *Drosophila*."). I understand the temptation of writing a bold title, but I think it is a bit misleading in the present case. I.e., it would be desirable that the title conveys the uncertainties of the data and their interpretation.

We agree with this suggestion. We have now amended this title to “Expansion and loss of sperm nuclear basic protein genes in *Drosophila* correspond with genetic conflicts between sex chromosomes.” However, we also want to highlight that de-masculinization of the X chromosome cannot explain the observed amplification and loss patterns of SNBP genes, except in cases of sex chromosome fusions. We now highlight the de-masculinization hypothesis for the latter case, but still strongly favor the genetic conflicts hypothesis.

2) "In contrast, we found no instances of pseudogenization or subsequent translocation to the X chromosome of SNBP genes that are still preserved on their original autosomal locations or involved in chromosome fusions between autosomes (0/16). This difference is highly significant (Figure 5 and Table S11; 3:5 versus 0:16, Fisher's exact test, P=0.03). " Readers should be warned that this pattern can also be explained by the well-known demasculinization of X chromosomes (e.g., Sturgill et al. Nature 2007, 450, 238-241)

We agree with this point and thank the reviewer for pointing this out. We now expressly raise the ‘de- masculinization of X chromosomes’ as one potential explanation of the pattern we observe here.

3) "Indeed, no meiotic drive has been documented in the montium species even though it is rampant in many other *Drosophila* lineages [38]." Two remarks here:a) the authors should make clear that they are referring to sex-chromosome meiotic drive.b) I think the evidence is much weaker than the sentence implies. Sex-chromosome meiotic drive is known in less than 20 *Drosophila* species, scattered throughout the phylogeny. As far as I know all cases were discovered by accident, so the sampling is biased towards model species (e.g., the obscura group, which was very popular around 1930-1960). So we do not know the true frequency of sex-ratio meiotic drive among *Drosophila* species, nor, say, if it is more common in the *Drosophila* or Sophophora species, if it is suspiciously absent in the montium group (as suggested by the authors), etc. I think these uncertainties should be acknowledged or, perhaps, given the weakness of the argument, the sentence should be deleted or attenuated.

We agree with this comment and have now removed this argument in our revision.

4) "X-Y chromosome fusions eliminate the extent of meiotic drive and may lead to the degeneration of otherwise conserved SNBP genes, whose functions as drive suppressors are no longer required. Thus, unlike in mammals, sex chromosome-associated meiotic drive appears to be the primary cause of SNBP evolutionary turnover in *Drosophila* species." The authors found that in the montium species the ancestral Y became incorporated into de X chromosome, and that montium species seem to have an inordinate amount of SNBP gene losses. They combine these two observations by suggesting that these SNBP became dispensable or deleterious because they originally were involved in XY meiotic drive. I think many readers will think that males in montium species are X/0, whereas in fact in all of them carry a Y chromosome (just, in most cases, more gene poor than "normal" Y-chromosomes). I do not think this is a fatal flaw for the explanation proposed by the authors, but certainly is a difficulty that should be acknowledged.

We agree with this point. It was not our intention to suggest that *montium* group males are X/O, but this could be misinterpreted as we originally stated. We now add a clarification that *montium* group males still harbor a Y chromosome, which is missing most ancestrally Y-linked genes.

Problems/suggestions with experiments and data analysis5) There is a section titled "CG30056 is universally retained in *Drosophila* but dispensable for male fertility in *D. melanogaster*". In this section and in the figures, it is stated, "Although CG30056 is the most conserved SNBP we surveyed, we found no clear difference in offspring number between heterozygous controls and homozygous knockout males (Figure 2B). (…) We found either no or weak evidence of fertility impairments in two different crosses with homozygous CG30056 knockout males.".I think the fertility data are weak for the purpose of the authors, and I strongly suspect that this conclusion is wrong. Let me explain why. At other passages of the manuscript, the authors classify the SNBP genes in three groups. (i) essential/important for male fertility: "Three genes (Mst77F, Prtl99C and ddbt) are essential for male fertility while knockdown or knockout of two other SNBP genes (ProtA, and ProtB) leads to significant reduction in male fertility [27-30, 32]." (ii) genes that do not appear to impair male fertility at all. (iii) untested. CG30056 was in the last group, and hence the authors produced knockouts, tested their effect in male fertility, and concluded that it belongs to the second group.Now, look at Figure 3B. The numbers of tested males are too small (it seems to range from 3 to 10), and male fertility is known to be a very noisy phenotype (as shown by the huge scatter in the authors' data). Furthermore, two different knockouts were tested, and both were nominally less fertile than the controls, and in one of them the difference is statistically significant. Taken at the face value, the knockouts seem to be perhaps ~25% less fertile than the controls. Another potentially big problem is that the "control males" actually carry visible dominant mutants (the balancers CyO or SM6) which certainly reduce their fitness, whereas the experimental males are wild-type for these mutants. Without the detrimental effect of these visible mutants in the controls, the difference to the CG30056 knockouts will probably be even larger. Note that the fertility effects of the genes ProtA, and ProtB (a.k.a. "Mst35B") , which the authors put in group "essential/important for male fertility" would not had been detected if assayed as the CG30056 gene: Tirmarche et al. (2014; the reference cited by the authors) stated that: "In fact, the impact of Mst35B on male fertility was only revealed when mutant males were allowed to mate with a large excess of virgin females (1 for 10; Figure 3F) but not with a 1:1 sex ratio (not shown). " The authors' fertility test did not used this type of challenge.My general impression is that the fertility effects of CG30056 may actually be similar to ProtA and ProtB. I think the authors should do a proper fertility test of CG30056, or remove this section. Another possibly useful approach would be to classify the SNBP genes in those essential for male fertility and those that are not essential, because "experimentally speaking" this is a safer distinction (e.g., the fertility testes reported by other authors may also had been quick tests). Since these genes only function in sperm and are under purifying selection (otherwise they would have been lost; also, all have dN/dS < 1 ), they all most likely affect male fertility to some extent. In case the section on male fertility stays, it will be necessary to provide more details. How many males were crossed for each genotype? In some cases in Figure 2B, it seems that as low as 3, but it may be data superposition in the graph. Please provide the raw data in the supplementary material.

We are very appreciative of the many important points raised by the reviewer. Rather than removing this conclusion, which is not central to our paper, we have now performed additional, well-controlled experiments to address the reviewer’s concerns, which we summarize below:

We hope these changes will satisfy the reviewer's concerns about this section of our paper.

6) "Our phylogenomic analyses also highlighted one *Drosophila* clade- the montium group of species (including D. kikkawai)- which suffered a precipitous loss of at least five SNBP genes that are otherwise conserved in sister and outgroup species (Figure 3). (…) Given our hypothesis that autosomal SNBP genes might be linked to the suppression of meiotic drive (above), we speculated that the loss of these genes in the montium group of *Drosophila* species may have coincided with reduced genetic conflicts between sex chromosomes in this clade." The montium data is an important part of the paper. I think the authors should test the statistical significance of this pattern.

We appreciate the reviewer’s suggestion. However, we are unable to perform the statistical tests suggested for technical reasons. We note that three loss events occurred in the ancestor of *D. montium* species, while two happened in the ancestors of most *D. montium* species. Since it’s hard to estimate the evolutionary rates using these internal branches, we can’t directly compare them to other branches using statistics. However, in response to the reviewer’s comments, we now more clearly contrast the fate of SNBPs between *D. montium* species and other *melanogaster* group species, noting that three of five genes lost in the *montium* group are retained in all other *melanogaster* group species.

Other points:7) "The five remaining SNBP genes (Mst33A, CG30056, CG31010, CG34269, and CG42355) remain cytologically uncharacterized [30]." I think it will be interesting if the authors look at other potentially useful resources: Vibranovski et al. papers which looked at gene expression in mitotic, meiotic and post-meiotic cells (https://mnlab.uchicago.edu/sppress/index.php), and the papers by several labs on testis single-cell transcriptomic data (Witt et al. 2021 PLOS Genetics. 17(8):e1009728 ; Nat Commun. 2021;12: 892). These may provide additional clues on the function of SNBP genes. There is also a recent report on sperm proteome (doi: https://doi.org/10.1101/2022.02.14.480191)

We are grateful to the reviewer for this suggestion. We now add the data from single-cell expression analyses from Witt et al. in Table 1—figure supplement 1. We found most SNBPs are expressed at late spermatocytes and early spermatids, although *CG30056* is primarily expressed in late spermatids, whereas *CG34269* is expressed earlier in late spermagonia. The data from Vibrranovski et al. also show similar patterns but don’t have four of these genes, including CG34269. The data from Mahadevaraju et al. are from larva testes, and lack some critical stages during spermatogenesis. Thus, we only report the data from Witt et al.

We also surveyed the proteome data as the reviewer suggested, but we only found 3 SNBPs (ProtA, ProtB, and Prtl99C) in the data. This did not include, Mst77F, which is the most highly expressed (see Table 2) and well-studied SNBP, so we suspect the proteomic study might be biased toward proteins from sperm tails.

Therefore, we decide not to include this analysis.

8) "Our inability to detect homologs beyond the reported species does not appear to result from their rapid sequence evolution. Indeed, abSENSE analyses [45] support the finding that Prtl99C, Mst77F, Mst33A, Tpl94 and CG42355 were recently acquired in Sophophora within 40 MYA. For example, the probability of a true homolog being undetected for Prtl99C and Mst77F is 0.07 and 0.18 (using E- value=1), respectively (Table S1, Methods)." This should be complemented by synteny analysis.

It may not have been clear from our original version that we did perform synteny analyses for all SNBP genes. We have now restated this more clearly in our revision.

9) I found the following sentence unclear: "However, we could only ascribe a sex chromosomal linked location for species if no data was available from either BUSCO genes or females (only males and mixed-sex flies)."

We modified the sentence to make it clearer: “However, we could not ascribe a sex-chromosomal linked location of a contig to either the X or Y chromosome in cases where there was no linkage information from BUSCO genes and no read data available from females, only from males and mixed-sex flies.”

10) "Using the available assemblies with Illumina-based chromosome assignment, we surprisingly found that most ancestrally Y-linked genes are not linked to autosomes as was previously suggested [by Dupim et al. 2018] (Figure 6A)."The new result of X-linkage is exciting, but the sentence is not exact: Dupim et al. 2018 made clear that they could only separate X/A from Y-linkage. E.g., the legend of their Figure 3: "Phylogeny and gene content of the Y chromosome in the montium subgroup. "M" means amplification only in males (i.e., Y- linkage), whereas "MF" means amplification in both sexes (autosomal or X-linkage)."

We are grateful to the reviewer for this correction. We now modified the sentence to make clear that Dupim et al. had “showed that many ancestrally Y-linked genes are present in females because of possible relocation to other chromosomes in the montium group.”

11) "The most parsimonious explanation for these findings is a single translocation of most of the Y chromosome to the X chromosome via a chromosome fusion in the ancestor of the montium group of species. Afterward, some of these genes relocated back to the Y chromosome in some species (Figure S6; Supplementary text)." Explanations for this pattern of "return to the Y" have been extensively discussed and tested in Dupim et al. 2008 (see their section "Why genes seem to return to the Y chromosome after Y incorporations?" ) The available evidence strongly suggests that it is not a case of relocation to the Y.

We thank the reviewer for raising this point. However, our conclusions disagree slightly with those from Dupim et al. 2018, in part because of additional sequencing in this clade. Dupim et al. suggested the possibility that most Y chromosomal loci duplicated to other chromosomes in the ancestor of the *D. montium* clade, following which each species degenerated either Y-linked or autosomal copies of genes. If this was the case, Y-linked copies should have diverged from X-linked copies since the ancestor of the *D. montium* clade. In contrast to this expectation, our phylogenetic analyses found that *D. kikkawai* Y-linked *PRY* is more closely related to X- linked *PRY* in all other related species (Figure 6—figure supplement 1). This result is much less parsimoniously explained by the ancient duplication event proposed by Dupim et al. and is more consistent with a ‘return-to-Y’ that we propose. We also make clear that, unlike *PRY*, we can’t differentiate the two hypotheses in the case of *kl-2*.

12) Figure 6B suggests that the authors assembled the "translocated Y" in D. triauraria. However, no direct data or account for this assembly is provided. Please clarify.

This was not our assembly. We searched all publicly available assemblies in the *montium* group and found one assembly (NCBI accession GCA_014170315.2) that assembled all ancestral Y-linked regions. We now clarify this in our revision.

13) "Why would meiotic drive only influence *Drosophila*, but not mammalian, SNBP evolution? One important distinction may arise from the timing of SNBP transcription. In *D. melanogaster*, SNBP genes are transcribed before meiosis but translated after meiosis [29, 43, 57]. Thus, SNBP transcripts from a single allele, e.g., Xlinked allele, are inherited and translated by all sperm, regardless of which chromosomes they carry. Consequently, they can act as meiotic drivers by causing chromatin dysfunction in sperm without the allele, e.g., Y-bearing sperm." During spermatogenesis *Drosophila* haploid cells actually are syncytial, which has interesting consequences for the evolution of male genes (Raices et al., Genome Res. 1115-1122, 2019). This may be relevant for the present paper.

We thank the reviewer for this suggestion. We now gratefully include this citation in our revision.

Reviewer #2 (Significance (Required)):See aboveReviewer #3 (Evidence, reproducibility and clarity (Required)):This manuscript by Chang and Malik consider the evolution of HMG-box-containing sperm nuclear basic proteins (SNBPs) across *Drosophila* species in phylogenetic context. Previous work in mammals had highlighted fast evolution of proteins involved in chromatin remodeling during spermatogenesis. Here, the authors provide evidence for widespread positive selection and likely involvement in genetic conflict in a set of proteins with analogous functions in *Drosophila*. Amongst other findings, the authors highlight biased amplification of SNBP paralogs on sex chromosomes along several *Drosophila* lineages, a tendency towards loss/pseudogenization following translocation onto a sex chromosome, and an intriguing concerted SNBP loss event in the montium group where parts of the Y chromosome have become fused to the X, thus nullifying the chance that genetic conflicts can play out via distorted segregation of sex chromosomes. The authors suggest that, taken together, their findings support widespread of SNBPs involvement (as instigators and repressors) in meiotic drive. Overall, I found the manuscript to be well written and thorough in its exploration of the evolutionary dynamics of SNBPs in this clade.

We thank the reviewer for the accurate summary and the kind comments.

Below, I have highlighted some aspects that I think would benefit from further attention, none of them major.Following their exploration of patterns of SNBP evolution in *Drosophila*, the authors highlight support of their data for genetic conflict between sex chromosomes. They also rightly acknowledge that other evolutionary drivers such as sperm competition might also play a role in, for example, fast evolution of certain SNBPs. Yet those (not mutually exclusive) alternatives are never pitted directly against each other. The focus is firmly on exploring the support for the sex chromosome genetic conflict model. Given that the authors highlight *Drosophila* as a great model in part because of its well characterized sperm biology (including comparative morphology), I wondered why the authors had not made an explicit attempt to see if SNBP evolution covaries with aspects of sperm morphology across *Drosophila*.

We do agree with the reviewer that it will be very interesting to test whether SNBP evolution covaries with sperm morphology in *Drosophila*. However, data on sperm morphology is scant in most *Drosophila* species. Indeed, this trait has only been well studied in clades with heteromorphic (different-sized) sperm but we agree this will be an exciting topic to consider in the future.

We also clarify better in our revised discussion that our analyses do not rule out a role for sperm competition or sperm morphology in driving the evolution of at least some SNBP genes. We note that a subset of SNBP genes undergo gene amplifications and loss, but most SNBP genes evolve rapidly including in species with gene loss. Thus, the meiotic drive hypothesis is not to the exclusion of other hypotheses.

The most intriguing part of the manuscript for me was the exploration of SNBP fate in the montium group, where the authors find evidence for an ancestral fusion event between the X and parts of the Y chromosome. The loss of SNBPs is certainly consistent with the conflict model but I was wondering to what extent this lineage is characterized more broadly by unusual evolution at the chromosomal level. Is there simply a lot of upheaval in montium, with more frequent gain/loss across the board? How specific is SNBP loss in the context of other orthologous groups? This could be investigated by looking at retention of other genes in other orthologous groups (in montium and some other control group) or perhaps by looking at synteny conservation.

This is a good suggestion. Using the same methodology as used in this paper, we found that very few *D. melanogaster* essential genes (<3% of >2000) are lost in any single species we surveyed here (unpublished data). However, we have not carried out similar analyses for all genes; given vastly different rates of evolution, this would be a significant undertaking. Thus, we are not able to make a direct comparison between SNBP genes and a control group, that would include other testes-specific or fertility-essential genes. Instead, we highlight the fact that since we identify SNBPs using syntenic analyses, we have known that the neighboring genes of SNBPs are much better conserved than the SNBP genes themselves in the *montium* group species.

In introducing SNBPs, the authors focus on their role as packaging agents. Clearly, SNBPs do package the genome in the sense that they bind to DNA and lead to reduced chromosome volume. But is this all packaging for packaging's sake (as portrayed by the sperm shape hypothesis)? Or is the situation a bit more nuanced, where condensation leads to a reduction of volume but also to a shutdown of transcription, protection from DNA damage, etc.? I think the focus on packaging alone is somewhat limiting when it comes to imagining how these proteins might act in the context of genomic conflicts. The authors may want to broaden their description of SNBPs in the Introduction accordingly.

We completely agree with the reviewer and are currently exploring these possibilities in follow-up studies on SNBP function. However, it is fair to add that this hypothesis has not been well-recognized, and we, therefore, prefer to include it in our revised Discussion rather than Introduction. However, we also think that SNBP packaging function might be targeted by Wolbachia-encoded toxins, speeding up their evolution (revised Discussion). We think there are many molecular possibilities for SNBPs.

The authors highlight that some SNBPs are expressed in mature sperm whereas others are transition proteins. The evidence for positive selection chiefly comes from the latter group (and "undefined" proteins that could also be transition proteins). Can the authors comment on whether this is expected/unexpected? Along the same lines, the authors highlight differences between *Drosophila* and mammals when it comes to the timing of transcription/translation during meiosis, suggesting that meiotic drive can happen in *Drosophila* because alleles are expressed early and can exert an effect after meiosis regardless of whether the associated locus is present in the gamete. I wonder how this relates, if at all, to the author's finding that transition SNBPs are more likely to be part of conflicts (as indicated by positive selection signals) compared to SNBPs in mature sperm.

We thank the reviewer for this comment. We expect that many genes expressed explicitly in spermatogenesis, including SNBP genes, would be under position selection, regardless of whether they are associated with X-Y conflicts. The positive selection signals could come from either X-Y conflicts, sperm competition, or conflicts with Wolbachia; we now discuss all of these in the Discussion.

In contrast, the amplification and loss of a subset of *Drosophila* SNBPs are more likely associated with X-Y conflicts. We note that known SNBPs retained in mature sperm are more likely to be subject to amplification than known transition proteins.

Regarding the timing of expression, it is true that transition SNBPs act earlier in spermatogenesis than SNBPs retained in mature sperm. However, for the meiotic drive hypothesis to apply, all it requires is for SNBP expression to precede sperm individualization, which it does for most SNBPs, including transition proteins.

It is not entirely clear from the text (and also e.g. Table S4) how dN and dS (and subsequently dN/dS) where calculated. I presume as a single estimate across the whole phylogeny? If so, how heterogeneous is dN/dS across the phylogeny and can the authors identify specific branches on which selective regimes are different? A branch-level analysis should be better powered than the site-level analysis the authors present, which requires repeated selection on the same set of sites to get a strong enough signal. A branch-specific assessment of evolution would be particularly valuable in combination when combined with the assessment of amplifications/losses.

We thank the reviewer for this question. The reviewer is correct. We estimated dN and dS in Supplementary file 4 across the whole phylogeny. We conducted branch tests for the amplification of tHMG only in the Dsim clade (Supplementary file 11).

We are interested in how SNBP amplification happened across species, but we need better gene annotation for their structure in many of these 19 independent cases. Moreover, we hope to combine these with transcriptomic analyses with detailed sequence analyses to reveal how the event happened and how gene conversion, gene duplication, and mutations affect their evolution. Each of these analyses requires extensive additional resources and analyses, and we feel are beyond the scope of this current paper.

The authors suggest that young SNBPs are more likely to encode essential, non-redundant male fertility functions (p7, third paragraph). I'm not sure whether this generalization is appropriate given the small sample. Tpl94D is as young as Mst77F/Prtl99C, tHMG and CG14835 homologs have been lost along different lineages and most of the events are in a single lineage leading up to D. kikkawai. Do the authors really feel that this generalization is warranted?

We agree with the reviewer. However, it is striking that the known fertility essential genes are either young or not universally conserved. We have therefore reworded our conclusion to make this contrast more accurate.

How do the sex-chromosomal amplifications differ in sequence from the ancestral autosomal copies? The authors suggest that the sex chromosomal copies might be involved in meiotic drive? Does the sequence offer a function as to how? (e.g. loss of charged residues/DNA-binding capacity)?

These are good questions. We do not know mechanistically how the sex-chromosome amplifications may cause meiotic drive. We did not observe the loss of positive charge or HMG domain in most sex-chromosomal amplified copies (Supplementary file 3). Our current working hypothesis is that they compete for the DNA binding with autosomal SNBP, and might interact with other proteins, e.g., heterochromatin proteins, to disturb sperm function. How they might function to cause meiotic drive is an active area of investigation in our and other labs.

I think it would be nice to have a final table/figure to summarizing the different lines of evidence for all the genes in Table 1 (i.e. positive selection yes/no, amplification in some lineages yes/no, sex chromosome translocations yes/no), for different lineages, including whether any of the HMG-box genes are unlikely to act as SNBPs.

We agree with this suggestion. We have now significantly revised and added to Table 2 to include this added information.

The evidence the authors present is often consistent with genetic conflicts between sex chromosomes. Is it cogent? Arguably not since direct tests of the mechanism are provided. I would therefore suggest a more cautious title than one stating that conflicts drive expansion and loss of SNBPs.

We agree with all three reviewers and have amended our title to highlight the correlation. We also discuss other possibilities that can drive SNBP evolution in our revised Discussion.

Typographical errors etc.:– P3. First paragraph: "One of the driving forces.… " I found this sentence a bit odd in terms of causality (changes in composition being portrayed as a force that leads to selection)

We thank the reviewer for pointing out the confusing construction. We modified the sentence to “The positive selection of SNBPs results in changes to their amino acid composition.”

– P3. Second paragraph: should be "HMG-box" rather than "HMB-box"

Fixed.

– P3. Fourth paragraph "…, consistent with the observation in mammals". I think "consistent" should be reserved for two observations that speak to the same phenomenon. SNBPs could evolve with no evidence for positive selection in *Drosophila* and that wouldn't exactly be "inconsistent" with mammals. It would just be different.

Fixed. We changed “consistent with” to “similar to”.

– P5. Fifth paragraph: should be "in the PAML package" rather than "in PAML package"

Fixed.

– P9. Second paragraph: "… montium group (Figure 5A…)" should be Figure 6A.

Fixed.

Cross-Consultation CommentsI have not much to add. The other reviews seem fair and well-informed from my somewhat-outside perspective. I don't know how tricky/time-consuming the suggested additional fly mating experiments are but want to note that, in general, I'm loath to "punish" authors of principally bioinformatic work for including some experiments. If experimental shortcomings can be addressed with appropriate caveats, that should be an option, as should removal of experimental data that – by the experts – would be considered too preliminary.

We thank the reviewer for their support. However, we felt that improved experiments on CG30056 role in fertility could broaden the scope of this paper, despite the additional time and labor commitment. We have now finished these experiments and they do reinforce our original conclusions with much greater support.

Reviewer #3 (Significance (Required)):I'm not enough of an expert in the field of SNBPs to assess the level of advance provided by this study.